# Stochastic Gradient Descent in Correlated Settings: A Study on Gaussian Processes

**Hao Chen**\*
University of Wisconsin-Madison
haochen@stat.wisc.edu

**Lili Zheng**\*
University of Wisconsin-Madison
lilizheng@stat.wisc.edu

**Raed Al Kontar**
University of Michigan
alkontar@umich.edu

**Garvesh Raskutti**
University of Wisconsin-Madison
raskutti@stat.wisc.edu

## Abstract

Stochastic gradient descent (SGD) and its variants have established themselves as the go-to algorithms for large-scale machine learning problems with independent samples due to their generalization performance and intrinsic computational advantage. However, the fact that the stochastic gradient is a biased estimator of the full gradient with correlated samples has led to the lack of theoretical understanding of how SGD behaves under correlated settings and hindered its use in such cases. In this paper, we focus on the Gaussian process (GP) and take a step forward towards breaking the barrier by proving minibatch SGD converges to a critical point of the full loss function and recovers model hyperparameters with rate $O(\frac{1}{K})$ up to a statistical error term depending on the minibatch size. Numerical studies on both simulated and real datasets demonstrate that minibatch SGD has better generalization over state-of-the-art GP methods while reducing the computational burden and opening up a new, previously unexplored, data size regime for GPs.

## 1 Introduction

The Gaussian process (GP) has seen many success stories in various domains, be it in optimization [42, 32], reinforcement learning [33, 20], time series analysis [19, 1], control theory [17, 23] and simulation meta-modeling [44, 26]. One can attribute such success to its natural Bayesian interpretation, uncertainty quantification capability and highly flexible model priors. Yet its main limitation is the $O(n^3)$ computation and $O(n^2)$ storage for $n$ training points [29]. Indeed, as mentioned in [13], a traditional large dataset for a GP is one with a few thousand data points and even those often require approximation techniques.

As a result, during the past two decades, a large proportion of papers on GPs tackled approximate inference procedures to reduce the computational demands and numerical instabilities (mainly due to the need for matrix inversions). This push towards scalability dates back to the seminal paper by Quiñonero-Candela and Rasmussen [27] in 2005 which unified previous approximation methods into a single probabilistic framework based on inducing points. Since then, many new methods have also been introduced. Most notable are: variational inference procedures that laid the theoretical foundation for the class of inducing point methods [8, 25, 43, 3, 40], mixture of experts models [9, 36], covariance tapering [11, 16] and kernel expansions [21, 28, 41]. On the other hand, there has been a recent push to utilize increasing computational power and GPU acceleration to solve exact GPs. This recent literature inlcudes distributed Cholesky factorizations [24], preconditioned

---

conjugate gradients (PCG) to solve linear systems [12] and kernel matrix partitioning to perform all matrix-vector multiplications [38]. Interestingly [38] was able to fit a bit more than 1 million data points using 8 GPUs in a few days.

One possible solution to extend GPs far beyond what is currently possible is through stochastic gradient decent (SGD) and its variants: drawing $m << n$ samples at each iteration and updating model parameters following the gradient of the loss function on the $m$ subsamples. Indeed, SGD, or more generally the capability of inference via minibatches (possibly also with second order information), has been a key propeller behind the success of deep learning in its various forms [22]. *The caveat however is that, unlike empirical loss minimization, there exists correlation across all samples where any finite collection of the samples has a joint Gaussian distribution with covariance characterized by an empirical kernel matrix.* This translates to the stochastic gradient being a biased estimator of the full gradient when taking expectation with respect to the random sampling. The lack of theoretical backing and understanding of how SGD behaves in such settings has long stood in the way of the use of SGD to do inference in GPs [13] and even in most correlated settings.

In this paper, we establish convergence guarantees for both the full gradient and the model parameters. Interestingly, without convexity or even Liptchitz conditions on the loss function, the structure of the GP leads to an optimization error term $O(\frac{1}{K})$ for both converging to a critical point and recovering true key parameters: noise variance and signal variance multiplied by the kernel function. Our proof takes two steps: first we concentrate the stochastic gradient to its conditional expectation using an $\epsilon$-net argument and then we show that the latter satisfies a strongly convex-like property by exploiting eigenvalues of the empirical kernel matrix. The proof and key findings offer standalone value beyond GPs and we hope they encourage researchers to further investigate SGD in other correlated settings such as Lévy, Itô and Markov processes.

Most importantly, however, the results open a new data size regime to explore GPs. We were able to train $n \approx 1.2 \times 10^6$ data points using a single CPU core in around 30 minutes. Recall, it took the most recent advancements in exact GPs a couple of days using 8 GPUs when $n \approx 10^6$ and $n$ is limited to approximately $10^4$ without GPU. We find that GPs inferred using SGD offer remarkably better performance in various case studies with different dataset sizes, noise levels and input dimensions. These results highlight the value of intrinsic regularization offered by SGD and also shed light on the value of increased data sizes in Bayesian non-parametric representations. We first start by listing our key findings in Section 1.1. We also note that the detailed proof is deferred to the appendix and only an outline is provided in Section 4.

## 1.1 Key Findings

We establish convergence guarantees for the minibatch SGD algorithm for training GP, sampling with or without replacement. Under regularity conditions, our results suggest the following:

- For a large enough minibatch size $m$, minibatch SGD converges to a critical point of the full log-likelihood loss function, and recovers the true hyperparameters, including the noise variance and the signal variance. To be specific, the full gradient and the estimation error of the hyperparameters evaluated at the $K$th iterate are bounded by an optimization error term $O(\frac{1}{K})$ and a statistical error term: $O(m^{-\frac{1}{2}})$ for the full gradient and the noise variance, and $O((\log m)^{-\frac{1}{2}})$ for the signal variance if the kernel function has exponential eigendecay, see Theorems 3.1 and 3.2.

- To guarantee the $O(\frac{1}{K})$ optimization error bound, no convexity or even Liptchitz condition on the loss function are assumed. Instead, we prove that the conditional expectation of the loss function given covariates $\mathbf{X}_n$ satisfies a relaxed property of strong convexity (see Lemma 4.1), which provides more flexibility in the choice of initial parameters.

- Through benchmarking with state-of-the-art methods on various datasets we show that SGD offers great value from both computational and statistical perspectives. Computationally, we scale to dataset sizes previously unexplored in GPs in a fraction of time needed for competing methods. Meanwhile statistically, we find that the induced regularization imposed by SGD improves generalization in GPs, specifically in large data settings.

## 2 Problem Setup

**Notations** Vectors and matrices are denoted by boldface letters, e.g., $\mathbf{K}_n$, $\boldsymbol{\theta}$, except for the full gradient $\nabla\ell(\boldsymbol{\theta})$ and stochastic gradient $g(\boldsymbol{\theta})$. For any vector $\mathbf{u} \in \mathbb{R}^p$, $u_i$ denotes its $i$th entry, and $\|\mathbf{u}\|_2 = \left(\sum_{i=1}^p u_i^2\right)^{\frac{1}{2}}$ denotes its $\ell_2$ norm. For any square matrix $\mathbf{A}$, $\lambda_i(\mathbf{A})$ denotes its $i$th largest eigenvalue.

We consider the Gaussian process model

$$f \sim \mathcal{GP}(m(\cdot), c(\cdot,\cdot)), \quad \mathbf{x}_1,\ldots,\mathbf{x}_n \overset{\text{i.i.d.}}{\sim} \mathbb{P},$$
$$y_i = f(\mathbf{x}_i) + \epsilon_i, \quad \epsilon_i \overset{\text{i.i.d.}}{\sim} \mathcal{N}(0,\sigma_\epsilon^2), \quad 1 \le i \le n, \tag{1}$$

where $\mathbf{x}_i \in \mathcal{X} \subset \mathbb{R}^D$ is the input, $m(\cdot): \mathcal{X} \to \mathbb{R}$ is the prior mean function, $c(\cdot,\cdot): \mathcal{X} \times \mathcal{X} \to \mathbb{R}$ is the prior covariance function, and $\epsilon_i$ is the observational noise with variance $\sigma_\epsilon^2$. Without loss of generality, we consider constant 0 mean function. Let the prior covariance function $c(\cdot,\cdot) = \sigma_f^2 k(\cdot,\cdot)$ for some kernel function $k(\cdot,\cdot): \mathcal{X} \times \mathcal{X} \to \mathbb{R}$, where $\sigma_f^2$ is the signal variance. We observe data points $\{(\mathbf{x}_i, y_i)\}_{i=1}^n$ generated from (1) and organize them into $(\mathbf{X}_n, \mathbf{y}_n) = ((\mathbf{x}_1,\ldots,\mathbf{x}_n)^\top, (y_1,\ldots,y_n)^\top)$, from which we aim to learn the hyperparameters in order to predict outputs from new inputs based on the posterior process.

Denote by $\boldsymbol{\theta}^* = (\sigma_f^2, \sigma_\epsilon^2)^\top \in \mathbb{R}^2$ the hyperparameters to be determined, and for notational convenience, we may also use $\theta_1^*$ to denote $\sigma_f^2$ and $\theta_2^*$ to denote $\sigma_\epsilon^2$ in the following. One direct approach to estimate $\boldsymbol{\theta}^*$ is by applying gradient descent to minimize the scaled negative log marginal likelihood function

$$\ell(\boldsymbol{\theta}; \mathbf{X}_n, \mathbf{y}_n) = -\frac{1}{n}\log p(\mathbf{y}_n | \mathbf{X}_n, \boldsymbol{\theta})$$
$$= \frac{1}{2n}[\mathbf{y}_n^\top \mathbf{K}_n^{-1}(\boldsymbol{\theta})\mathbf{y}_n + \log|\mathbf{K}_n(\boldsymbol{\theta})| + n\log(2\pi)] \tag{2}$$

over $\boldsymbol{\theta} \in (0,\infty)^2$, where $\mathbf{K}_n(\boldsymbol{\theta}) = \theta_1 \mathbf{K}_{f,n} + \theta_2 \mathbf{I}_n \in \mathbb{R}^{n\times n}$ is the marginal covariance matrix for noisy observations $\mathbf{y}_n$ given $\mathbf{X}_n$, and $\mathbf{K}_{f,n} \in \mathbb{R}^{n\times n}$ is the kernel matrix of $k(\cdot,\cdot)$ evaluated at $\mathbf{X}_n$, i.e. $(\mathbf{K}_{f,n})_{i,j} = k(\mathbf{x}_i, \mathbf{x}_j)$. For notational convenience we will omit $\mathbf{K}_n(\boldsymbol{\theta})$ to $\mathbf{K}_n$ when $\boldsymbol{\theta}$ is clear from the context and denote $\mathbf{K}_n(\boldsymbol{\theta}^*)$ by $\mathbf{K}_n^*$. In this case, the derivative of $\ell(\boldsymbol{\theta})$ is of particular interest to us where each of its entries takes the form

$$(\nabla\ell(\boldsymbol{\theta}; \mathbf{X}_n, \mathbf{y}_n))_l = \frac{1}{2n}\left[-\mathbf{y}_n^\top \mathbf{K}_n^{-1}\frac{\partial \mathbf{K}_n}{\partial\theta_l}\mathbf{K}_n^{-1}\mathbf{y}_n + \text{tr}\left(\mathbf{K}_n^{-1}\frac{\partial\mathbf{K}_n}{\partial\theta_l}\right)\right]$$
$$= \frac{1}{2n}\text{tr}\left[(\mathbf{K}_n^{-1}(\mathbf{I}_n - \mathbf{y}_n\mathbf{y}_n^T\mathbf{K}_n^{-1})\frac{\partial\mathbf{K}_n}{\partial\theta_l}\right], \quad 1 \le l \le 2, \tag{3}$$

where $\theta_l$ is the $l$th element of $\boldsymbol{\theta}$ and $(\partial\mathbf{K}_n/\partial\theta_l)_{ij} = \partial(\mathbf{K}_n)_{ij}/\partial\theta_l$. For notational convenience we will suppress $\mathbf{X}_n, \mathbf{y}_n$ and use $\nabla\ell(\boldsymbol{\theta})$ instead. Notice that the computation in (3) is dominated by the calculation of $\mathbf{K}_n^{-1}$, which requires $O(n^3)$ time. In order to reduce the computational cost of training, we consider the minibatch stochastic gradient descent approach to optimize (2).

### 2.1 Minibatch SGD algorithm

Let $\xi$ be a random subset of $\{i\}_{i=1}^n$ of size $m$, then $\{(\mathbf{x}_i, y_i)\}_{i\in\xi}$ is the corresponding subset of data points which we organize into $(\mathbf{X}_\xi, \mathbf{y}_\xi)$, where $\mathbf{X}_\xi$ is the submatrix formed by the rows of $\mathbf{X}_n$, and $\mathbf{y}_\xi$ is the subvector of $\mathbf{y}_n$, both indexed by $\xi$. Define $g(\boldsymbol{\theta}; \mathbf{X}_\xi, \mathbf{y}_\xi) \in \mathbb{R}^2$ as an approximation to $\nabla\ell(\boldsymbol{\theta}; \mathbf{X}_n, \mathbf{y}_n)$ that can be calculated from this subset, i.e.,

$$(g(\boldsymbol{\theta}; \mathbf{X}_\xi, \mathbf{y}_\xi))_l = \frac{1}{2s_l(m)}\text{tr}\left[(\mathbf{K}_\xi^{-1}(\mathbf{I}_m - \mathbf{y}_\xi\mathbf{y}_\xi^\top\mathbf{K}_\xi^{-1})\frac{\partial\mathbf{K}_\xi}{\partial\theta_l}\right], \quad 1 \le l \le 2, \tag{4}$$

where $\mathbf{K}_\xi$ is the covariance matrix of $\mathbf{y}_\xi$ while also being the principle submatrix formed by the rows and columns of $\mathbf{K}_n$ indexed by $\xi$. A natural choice for $s_l(m)$ is $m$, but we will see in Section 3 that setting $s_1(m) \asymp \log m$ and $s_2(m) = m$ would lead $\theta_1^{(k)}$ and $\theta_2^{(k)}$ to both converge to the true hyperparameters. Algorithm 1 summarizes the steps of minibatch SGD, where we do not specify whether minibatches are sampled with or without replacement since our theoretical guarantees will hold true under both scenarios.

**Algorithm 1:** Minibatch SGD

---

**1** Input: $\boldsymbol{\theta}^{(0)} \in \mathbb{R}^2$, initial step size $\alpha_1 > 0$.

**2 for** $k = 1, 2, \ldots, K$ **do**

**3** $\quad$ Randomly sample a subset of indices $\xi_k$ of size $m$;

**4** $\quad$ Compute the stochastic gradient $g(\boldsymbol{\theta}^{(k)}; \mathbf{X}_{\xi_k}, \mathbf{y}_{\xi_k})$;

**5** $\quad$ $\alpha_k \leftarrow \frac{\alpha_1}{k}$;

**6** $\quad$ $\boldsymbol{\theta}^{(k)} \leftarrow \boldsymbol{\theta}^{(k-1)} - \alpha_k g(\boldsymbol{\theta}^{(k-1)}; \mathbf{X}_{\xi_k}, \mathbf{y}_{\xi_k})$;

**7 end for**

---

## 3 Theoretical Guarantees

In this section, we provide convergence guarantees for Algorithm 1, including error bounds for $\|\boldsymbol{\theta}^{(k)} - \boldsymbol{\theta}^*\|_2^2$ and $\nabla \ell(\boldsymbol{\theta}^{(k)})$. The following assumptions are needed for our theoretical results.

**Assumption 3.1** (Exponential eigendecay). *The eigenvalues of kernel function $k(\cdot, \cdot)$ w.r.t. probability measure $\mathbb{P}$ are $\{Ce^{-bj}\}_{j=0}^{\infty}$, where $C \leq 1$ is regarded as a constant.*

This exponential eigendecay assumption is satisified by the RBF kernels. In fact, for kernel functions with a different decay rate (e.g., polynomial decay), similar convergence guarantees shall still hold, except that the error bounds may scale differently w.r.t. the minibatch size $m$. The requirement $C \leq 1$ is only for theoretical convenience, and it suffices to have a bounded $C$.

**Assumption 3.2** (Bounded iterates). *Both $\boldsymbol{\theta}^*$ and $\boldsymbol{\theta}^{(k)}$ for $0 \leq k \leq K$ lie in $[\theta_{\min}, \theta_{\max}]^2$, where $0 < \theta_{\min} < \theta_{\max}$.*

**Assumption 3.3** (Bounded stochastic gradient). *For all $0 \leq k < K$, $\|g(\boldsymbol{\theta}^{(k)}; \mathbf{X}_{\xi_{k+1}}, \mathbf{y}_{\xi_{k+1}})\|_2 \leq G$ for some $G > 0$.*

The following theorem guarantees the convergence of the parameter iterates under these assumptions.

**Theorem 3.1** (Convergence of parameter iterates). *Under Assumptions 3.1 to 3.3, when $m > C$ for some constant $C > 0$, we have the following results under two corresponding conditions on $s_l(m)$:*

*1. If $s_2(m) = m$, $\frac{3}{2\gamma} \leq \alpha_1 \leq \frac{2}{\gamma}$ where $\gamma = \frac{1}{4\theta_{\max}^2}$, then for any $0 < \varepsilon < C\frac{\log\log m}{\log m}$, with probability at least $1 - CK\exp\{-cm^{2\varepsilon}\}$,*

$$(\theta_2^{(K)} - \theta_2^*)^2 \leq \frac{8G^2}{\gamma^2(K+1)} + Cm^{-\frac{1}{2}+\varepsilon}. \tag{5}$$

*2. If in addition to $s_2(m) = m$, $s_1(m)$ is set to $\tau \log m$ where $\tau > \frac{64\theta_{\max}^4}{b\theta_{\min}^4}$, $\frac{3}{2\gamma} \leq \alpha_1 \leq \frac{2}{\gamma}$ where $\gamma$ depends on $\tau$, then for any $0 < \varepsilon < \frac{1}{2}$, with probability at least $1 - CK\exp\{-c(\log m)^{2\varepsilon}\}$,*

$$\|\boldsymbol{\theta}^{(K)} - \boldsymbol{\theta}^*\|_2^2 \leq \frac{8G^2}{\gamma^2(K+1)} + C(\log m)^{-\frac{1}{2}+\varepsilon}. \tag{6}$$

*Here $c, C > 0$ depend only on $\theta_{\min}, \theta_{\max}, b$.*

**Remark 3.1.** *Theorem 3.1 suggests that the noise variance parameter $\theta_2^{(K)}$ is guaranteed to converge to the truth $\theta_2^*$, with the optimization error term $O(\frac{1}{K})$ and the statistical error term $O(m^{-\frac{1}{2}+\varepsilon})$ with high probability, if $\varepsilon \log m$ is large, the initial stepsize is appropriately chosen and $s_2(m) = m$. Furthermore, if we let $s_1(m) = \tau \log m$, then Algorithm 1 achieves convergence for both $\theta_1^{(K)}$ and $\theta_2^{(K)}$ with statistical error $O((\log m)^{-\frac{1}{2}+\varepsilon})$.*

**Remark 3.2.** *The optimization error $O(\frac{1}{K})$ is credited to the structure of the GP loss function, which satisfies a relaxation of strong convexity (details provided in Section 4). The different eigenvalue structures of $\mathbf{K}_{f,\xi}$ and $\mathbf{I}_m$ lead to different rates of statistical errors for $\theta_1^*$ and $\theta_2^*$, while the fact that statistical errors depend on $m$ instead of $n$ is due to the correlation among $\mathbf{y}_\xi$ from different minibatches.*

**Remark 3.3.** *For the second case where $s_1(m) = \tau \log m$, $\gamma$ needs to satisfy*

$$\gamma = \min\left\{\frac{1}{32\tau b\theta_{\max}^2}, \frac{1}{4\theta_{\max}^2} - \frac{2\theta_{\max}^2}{\tau b\theta_{\min}^4}\right\}, \tag{7}$$

**Remark 3.4.** *One possible extension to our current set-up is to assume the covariance function $c(\cdot, \cdot)$ to be the summation over multiple kernel functions: $c(\cdot, \cdot) = \sum_{l=1}^{M} \sigma_{f,l}^2 k_l(\cdot, \cdot)$ for $M > 1$. To establish convergence guarantees for this case, we can follow similar arguments of the current proof with the additional assumption that the kernel matrices for all kernels $k_1(\cdot, \cdot), \ldots, k_M(\cdot, \cdot)$ share the same eigenvectors, which facilitates the analysis for the gradient.*

Based on Theorem 3.1, we also derive the following convergence guarantee for the full gradient.

**Theorem 3.2** (Convergence of full gradient)**.** *Under Assumptions 3.1 to 3.3, if $\frac{3}{2\gamma} \leq \alpha_1 \leq \frac{2}{\gamma}$ for $\gamma = \frac{1}{4\theta_{\max}^2}$, $m > C$, $s_2(m) = m$, then for any $0 < \varepsilon < C\frac{\log\log m}{\log m}$, with probability at least $1 - CK\exp\{-cm^{2\varepsilon}\}$,*

$$\|\nabla\ell(\boldsymbol{\theta}^{(K)})\|_2^2 \leq C\left[\frac{G^2}{K+1} + m^{-\frac{1}{2}+\varepsilon}\right], \tag{8}$$

*holds, where $c, C > 0$ depend only on $\theta_{\min}, \theta_{\max}, b$.*

Theorem 3.2 implies that, running SGD for sufficiently many iterations with large minibatch size leads to the convergence to a critical point of $\ell(\boldsymbol{\theta})$.

# 4 Proof Overview

In this section, we present the proof overview for the first part of Theorem 3.1 and Theorem 3.2. The proof of the second part in Theorem 3.1 follows similar ideas although requiring more careful analysis. With a bit abuse of notation, we will omit $g(\boldsymbol{\theta}^{(k)}; \mathbf{X}_{\xi_{k+1}}, \mathbf{y}_{\xi_{k+1}})$ to $g(\boldsymbol{\theta}^{(k)})$ and denote its conditional expectation $\mathbb{E}(g(\boldsymbol{\theta}^{(k)})|\mathbf{X}^{\xi_{k+1}})$ by $g^*(\boldsymbol{\theta}^{(k)})$. Similarly we define $\nabla\ell^*(\boldsymbol{\theta}^{(k)}) = \mathbb{E}(\nabla\ell(\boldsymbol{\theta}^{(k)})|\mathbf{X}_n)$.

Due to the bias in the stochastic gradient, we take the followings steps instead of directly drawing the connection between $g(\boldsymbol{\theta}^{(k)})$ and $\nabla\ell(\boldsymbol{\theta}^{(k)})$:

- For proving the first part of Theorem 3.1:
    - We first show that the conditional expectation $g^*(\boldsymbol{\theta}^{(k)})$ of the stochastic gradient has a property similar to strong convexity, see Lemma 4.1.
    - We then prove that $g(\boldsymbol{\theta})$ is close to its conditional expectation $g^*(\boldsymbol{\theta})$ uniformly over all possible $\boldsymbol{\theta}$, and thus $g(\boldsymbol{\theta}^{(k)})$ is close to $g^*(\boldsymbol{\theta}^{(k)})$. Applying Lemma 4.2 to each minibatch leads to the desired result.

    These two steps lead to the $O(\frac{1}{K})$ optimization error rate for $(\theta_2^{(k)} - \theta_2^*)^2$, and a statistical error rate depending on $m$, as shown in Theorem 3.1.

- For proving Theorem 3.2:
    - Lemma 4.2 suggests that $\nabla\ell(\boldsymbol{\theta}^{(k)})$ is close to $\nabla\ell^*(\boldsymbol{\theta}^{(k)})$

    - The eigendecay of kernel matrices ensures that $\|\nabla\ell^*(\boldsymbol{\theta}^{(k)})\|_2$ is controlled by $(\theta_2^{(k)} - \theta_2^*)^2$, which is upper bounded in Theorem 3.1.

    These steps above provide us with the same error bound of $\|\nabla\ell^*(\boldsymbol{\theta}^{(k)})\|_2$ from that of $(\theta_2^{(k)} - \theta_2^*)^2$ in Theorem 3.1.

## 4.1 Key Lemmas

The following two lemmas are the key building blocks of the proof: one shows the nice convex-like property of $g^*(\boldsymbol{\theta}^{(k)})$, the other establishes a uniform bound for the statistical error $\nabla\ell(\boldsymbol{\theta}) - \nabla\ell^*(\boldsymbol{\theta})$ over $\boldsymbol{\theta} \in [\theta_{\min}, \theta_{\max}]^2$, and thus also bounds $g(\boldsymbol{\theta}^{(k)}) - g^*(\boldsymbol{\theta}^{(k)})$;

**Lemma 4.1** (Strongly convex-like property of $g^*(\boldsymbol{\theta}^{(k)})$). *Under Assumptions 3.1 to 3.3, if $s_2(m) = m$, $m > C$, then with probability at least $1 - 2Km^{-c}$, the following claim holds true for $0 \leq k < K$:*

$$(\theta_2^{(k)} - \theta_2^*)(g^*(\boldsymbol{\theta}^{(k)}))_2 \geq \frac{1}{8\theta_{\max}^2}(\theta_2^{(k)} - \theta_2^*)^2 - \frac{C \log m}{m}, \tag{9}$$

*Here $C > 0$ depends only on $\theta_{\min}, \theta_{\max}, b$.*

Lemma 4.1 is a relaxation of strong convexity, but leads to similar convergence guarantees from running SGD on strongly convex objectives. The approximate "curvature" parameter, $\frac{1}{8\theta_{\max}^2}$ on the R.H.S of (9), remains a constant regardless of how large $m$ is. To guarantee the constant "curvature", we establish novel upper and lower bounds on $\sum_{j=1}^m \lambda_j^l (\theta_1^{(k)} \lambda_j + \theta_2^{(k)})^{-2}$ with high probability when $m$ is large, where $\lambda_j$ is the $j$th largest eigenvalue of $\mathbf{K}_{f,n}$, $l = 0, 1, 2$. The proof is based on the established error bounds for the empirical eigenvalues in [6] and the eigendecay of the kernel $k(\cdot, \cdot)$.

**Lemma 4.2** (Uniform statistical error). *Under Assumption 3.1 to 3.3, for any $x > 0$, $1 \leq l \leq 2$, we have*

$$\mathbb{P}\left(\sup_{\boldsymbol{\theta} \in [\theta_{\min}, \theta_{\max}]^2} |(\nabla\ell(\boldsymbol{\theta}))_i - (\nabla\ell^*(\boldsymbol{\theta}))_i| > Cx\right) \leq \delta(x), \tag{10}$$

*where $\delta(x) \leq C(\log x)^4 \exp\{-cn \min\{x^2, x\}\}$. Here $c, C > 0$ only depend on $\theta_{\min}, \theta_{\max}, b$.*

The major difficulty in the proof of Lemma 4.2 is to control the error term uniformly over $\boldsymbol{\theta} \in [\theta_{\min}, \theta_{\max}]^2$. We need an uniform error bound, since $g^*(\boldsymbol{\theta}^{(k)})$ is no longer the conditional expectation of $g(\boldsymbol{\theta}^{(k)})$ if conditioning on the past iterate $\boldsymbol{\theta}^{(k)}$. Although the set $[\theta_{\min}, \theta_{\max}]^2$ has constant dimension, the kernel matrix $\mathbf{K}_n(\boldsymbol{\theta}) \in \mathbb{R}^{n \times n}$ is of high dimension and is determined by $\boldsymbol{\theta}$ in a non-linear way. Our solution is to explore the Taylor's expansion of $\nabla\ell(\boldsymbol{\theta}) - \nabla\ell^*(\boldsymbol{\theta})$, then use truncation and covering arguments.

## 5 Related Work

As mentioned earlier, there are several methods trying to tackle the computational complexity of GPs. Those can be roughly split into three categories, though it is by no means an exhaustive list (see the survey in [1]). **Exact inference via matrix vector multiplications (MVM):** This recent class of literature has had the most success in scaling GPs. Initially such approaches depended on a structured kernel matrix where data lies in a regularly spaced grid [30, 39]. Then with the help of GPU acceleration, conjugate gradient and distributed Cholesky factorization, MVMs were applied to more general settings [38, 12, 37]. Such approaches have training complexity of $O(n^2)$ ($O(n \log n)$ possible on spaced grids), yet amenable to distributed computation and GPU acceleration. **Sparse approximate inference:** This class of methods is based on a low rank approximation of the empirical kernel matrix where $\mathbf{K}_n \approx \mathbf{K}_{nz}\mathbf{K}_{zz}^{-1}\mathbf{K}_{zn}$ and $z$ denotes a set of inducing points with cardinality$(z) = n_z << n$ [18, 2, 8, 43, 31]. Their time complexity is mainly $O(n_z^2 n)$ which can be reduced to $O(n + cn_z)$ for structured and regularly spaced grids. Indeed, sparse GPs have gained increased attention since variational inference (VI) laid the theoretical foundation of this class of inducing points/kernel approximations (starting from the early work of Titsias [35]). **Stochastic variational inference (SVI):** Following the work of [15], SVI was introduced to GPs in [13]. The key idea is to introduce a variational distribution over the inducing points so that the VI framework is amenable to stochastic optimization. This leads to a complexity of $O(n_z^3)$ at each iteration [14, 5]. Unfortunately, recent results in [7] show the need for at least $O(\log^D n)$ inducing points for Gaussian kernels, which implies a superlinear growth with the input dimension.

## 6 Practical Considerations

### 6.1 Sampling Scheme

Apart from sampling uniform minibatches stated in Algorithm 1, one may also consider sampling nearby minibatches in practice. In our case studies, we demonstrate a particular nearby sampling strategy, i.e., nearest-neighbor search, where a minibatch consists of an uniformly sampled data point

and its $m-1$ nearest neighbors within the data pool. We may construct a $k$-$d$ tree to conduct such search, which finds the $m-1$ nearest neighbors for every data point in a given dataset of size $n$ in $O(n \log n)$ time and $O(n)$ space.

## 6.2 Optimizing lengthscale

In practice, when considering the RBF kernel, it is necessary to estimate the lengthscale parameters as well. Similar to Algorithm 1, we can update the lengthscale parameters alongside the variance parameters using minibatch SGD. Despite the challenge of providing theoretical guarantees for lengthscales, our numerical experiments utilize minibatch SGD to estimates all parameters (including lengthscales) and yield superior results over state-of-the-art methods on a wide range of datasets.

## 6.3 Prediction

Although our main focus in this paper is estimating the hyperparameters, the last step when applying GP in real applications is always prediction. With the estimated optimal hyperparameters, various strategies can be applied to calculate the predictive mean for $\mathbf{x}_*$ and the predictive covariance between $\mathbf{x}_*$ and $\mathbf{x}'_*$, following the posterior, i.e.,

$$m_{\text{post}}(\mathbf{x}_*) = \sigma_f^2 \mathbf{k}_{\mathbf{X}_n \mathbf{x}_*}^\top \mathbf{K}_n^{-1} \mathbf{y}_n, \qquad c_{\text{post}}(\mathbf{x}_*, \mathbf{x}'_*) = \sigma_f^2 k(\mathbf{x}_*, \mathbf{x}'_*) - \sigma_f^4 \mathbf{k}_{\mathbf{X}_n \mathbf{x}_*}^\top \mathbf{K}_n^{-1} \mathbf{k}_{\mathbf{X}_n \mathbf{x}'_*}, \quad (11)$$

where $\mathbf{k}_{\mathbf{X}_n \mathbf{x}_*} = (k(\mathbf{x}_1, \mathbf{x}_*) \ldots, k(\mathbf{x}_n, \mathbf{x}_*))^\top$. The main computational cost comes from the linear solvers in (11). In general, for $n < 10^4$, they can be computed via Cholesky decomposition; for $n < 10^5$, preconditioned conjugate gradient (PCG) [12] can be applied for acceleration; for $n < 10^6$, PCG with partitioned kernel [38] could provide further speed up, if distributed computational resources are available. Another practical but less ideal strategy when predicting with extremely large $n$ is to utilize $n_1$ nearest neighboring data points of $\mathbf{x}_*$ to approximate $m_{\text{post}}(\cdot)$, where $n_1 < n$ is determined by computational resource. This is due to the interpolation nature of Gaussian process prediction.

# 7 Numerical Results

## 7.1 Numerical Illustration of Theory

In this section, we conduct simulation studies to verify our theoretical results[†]. We consider $n = 1,024$, $\mathbf{x}_i \overset{i.i.d.}{\sim} \mathcal{N}(0, 5^2)$ and $\mathbf{y}_n \sim \mathcal{N}(\mathbf{0}, \sigma_f^2 \mathbf{K}_{f,n} + \sigma_\epsilon^2 \mathbf{I}_n)$. $\mathbf{K}_{f,n}$ is an RBF kernel matrix with known lengthscale $l = 0.5$. The underlying true parameters are $\sigma_f^2 = 4$ and $\sigma_\epsilon^2 = 1$. In each experiment, we perform 25 epochs of minibatch SGD updates with diminishing step sizes $\alpha_k = \alpha_1/k$. Notice that similar numerical results can be obtained by sampling minibatches with replacement. We let scaling factors $s_1(m) = 3 \log m$ for $\sigma_f^2$ and $s_2(m) = m$ for $\sigma_n^2$. Each experiment is repeated 10 times with independent data pools.

Fig. 1 shows the convergence of parameters. First of all, the curves exhibit $O(\frac{1}{K})$ convergence rate stated in Theorem 3.1. In addition, the convergence points of $\sigma_\epsilon^2$ are significantly more concentrated around its truth than that of $\sigma_f^2$, which is consistent with the $O((\log m)^{-\frac{1}{2}})$ statistical error for $\sigma_f^2$ and $O(m^{-\frac{1}{2}})$ statistical error for $\sigma_\epsilon^2$ stated in Theorem 3.1.

Fig. 2 displays the effect of minibatch size $m$ on the convergence of the full gradient. As we can see, the curves flatten slower and become less concentrated as minibatch size decreases, suggesting that larger minibatch size results in faster convergence and smaller statistical error for the full gradient. Additionally, the convergence points of $\log(||\nabla \ell(\boldsymbol{\theta}^{(k)})||_2^2)$ scale linearly with minibatch size $m$, indicating $O(m^{-\frac{1}{2}})$ statistical error for $||\nabla \ell(\boldsymbol{\theta}^{(k)})||_2^2$. The above observations confirm our statements in Theorem 3.2. Due to space limit, we defer the figures demonstrating the effect of $m$ on the convergence of parameters to the supplementary file, which also supports our theoretical results.

---

[†]The R functions for conducting numerical experiments are available online: https://github.com/UMDataScienceLab/SGD-in-Gaussain-processes.

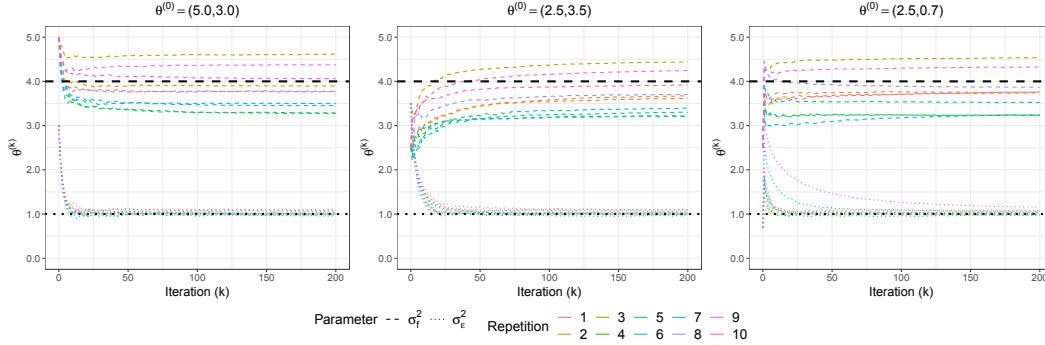

Figure 1: Illustration of the convergence of parameters from different initial points with $m = 128$. Lines in black denote the true parameters. The respective initial step sizes $\alpha_1$ are $9, 9$, and $6$.

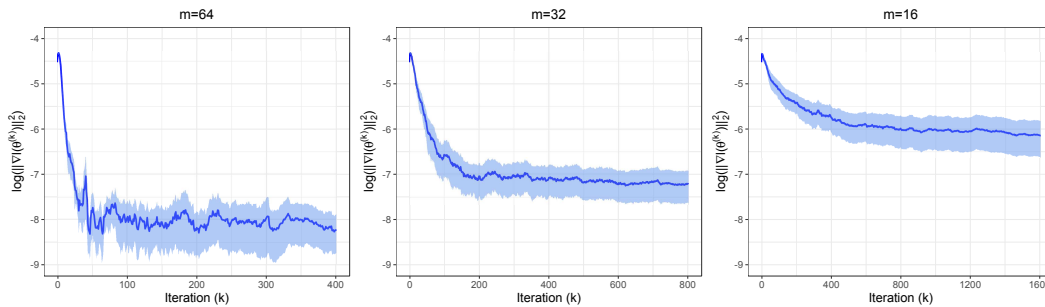

Figure 2: Comparison of the convergence of the full gradient with varying minibatch sizes. The mean of $\|\nabla\ell(\boldsymbol{\theta}^{(k)})\|_2^2$ and the region within its one standard error over 10 repetitions are shown in log scale. The three experiments share initial point $\boldsymbol{\theta}^{(0)} = (5.0, 3.0)$ and inital step size $\alpha_1 = 9$.

## 7.2 Case Studies

In this section, we first compare the generalization performance and training time of our stochastic gradient-based GP (sgGP) with PCG-based exact GP (EGP) [38], sparse GP regression (SGPR) [35] and stochastic variational GP (SVGP) [13] on various benchmark datasets. Notice that our approach differs from EGP in model selection but shares the same formula in prediction. We then demonstrate sgGP on toy datasets of size $2 \times 10^6$. The real datasets are from UCI repository [10] and the simulated datasets are from Virtual Library of Simulation Experiments [34].

Throughout all experiments, we consider constant 0 prior mean function and scaled RBF kernel prior covariance function with a separate lengthscale for each input dimension. Therefore, the hyperparameters to be learned are lengthscales, signal variance and noise variance. We conduct 10 independent trials on each dataset. In each trial, we randomly split the dataset into 60% training set and 40% test set. In addition, the training set is normalized to 0 mean and 1 standard deviation, and the test set is scaled accordingly.

During training (model selection), the hyperparameters and variational parameters are learned through minimizing the negative log marginal likelihood or its surrogate. We follow similar setups in [38]. For sgGP, we apply nearest-neighbor sampling strategy and perform 100 epochs of Adam with minibatch size $m = 16$ and a learning rate of $0.01$. For EGP, we perform 100 iterations of Adam with a learning rate of $0.1$. For SGPR and SVGP, we set the number of inducing points to $512$ and $1,024$, respectively, following theoretical recommendations in [7]. We carry out 100 iterations of Adam with a learning rate of $0.1$ for SGPR and 100 epochs of Adam with minibatch size $m = 1,024$ for SVGP. For the fairness of comparison, we do not perform any pretraining or fine-tuning, and we ensure different methods share a random but common starting point in each trial.

We code the training of sgGP with base R functions and use R package RANN [4] for nearest-neighbor search. The prediction of sgGP, together with the training and prediction of EGP, SGPR and SVGP are implemented through GPyTorch [12, 38]. Each experiment is performed on a single core of Intel

Table 1: Comparison of root-mean-square-error (RMSE) and training time of different GPs on benchmark datasets. We report the mean and standard error of RMSE as well as the mean and standard deviation of training time over 10 trials. The best results are in bold (lower is better). For query and borehole datasets, we are unable to fit with EGP due to memory limit.

| Dataset | Size | $D$ | RMSE | | | | Training Time (min) | | | |
|---|---|---|---|---|---|---|---|---|---|---|
| | | | sgGP | EGP | SGPR | SVGP | sgGP | EGP | SGPR | SVGP |
| Levy | 10,000 | 4 | **0.265** ± 0.003 | 0.312 ± 0.003 | 0.564 ± 0.010 | 0.582 ± 0.013 | **0.51** ± 0.00 | 11.48 ± 1.28 | 4.04 ± 0.51 | 14.58 ± 0.07 |
| Griewank | 10,000 | 6 | **0.071** ± 0.000 | 0.185 ± 0.073 | 0.132 ± 0.003 | 0.093 ± 0.005 | **0.61** ± 0.01 | 15.25 ± 3.72 | 1.93 ± 0.31 | 13.18 ± 0.58 |
| Bike | 17,379 | 17 | **0.221** ± 0.002 | 0.228 ± 0.002 | 0.276 ± 0.004 | 0.250 ± 0.010 | **1.98** ± 0.03 | 31.48 ± 7.45 | 5.31 ± 2.05 | 25.26 ± 3.97 |
| Energy | 19,735 | 27 | **0.786** ± 0.001 | 0.802 ± 0.007 | 0.843 ± 0.006 | 0.795 ± 0.005 | **3.15** ± 0.04 | 54.39 ± 8.01 | 5.41 ± 0.73 | 25.09 ± 5.50 |
| PM2.5 | 41,757 | 15 | 0.287 ± 0.002 | **0.286** ± 0.003 | 0.638 ± 0.005 | 0.540 ± 0.010 | **5.21** ± 0.04 | 385.51 ± 42.59 | 13.59 ± 2.30 | 52.46 ± 10.08 |
| Protein | 45,730 | 9 | **0.663** ± 0.006 | 0.694 ± 0.004 | 0.715 ± 0.003 | 0.676 ± 0.004 | **3.40** ± 0.03 | 500.33 ± 65.62 | 19.55 ± 1.66 | 55.27 ± 13.09 |
| Query | 100,000 | 4 | **0.053** ± 0.000 | — | 0.058 ± 0.002 | 0.061 ± 0.000 | **6.40** ± 0.10 | — | 20.73 ± 1.63 | 124.73 ± 22.25 |
| Borehole | 1,000,000 | 8 | **0.172** ± 0.000 | — | 0.176 ± 0.000 | 0.173 ± 0.000 | **67.29** ± 13.39 | — | 857.60 ± 76.02 | 1380.86 ± 11.32 |

Table 2: Illustration of sgGP on toy datasets. We follow similar setups in Table 1 but train 25 epochs.

| **Dataset** | Size | $D$ | **RMSE** | **Training Time** (min) | **Memory Usage** (GB) |
|---|---|---|---|---|---|
| OTL Circuit | 2,000,000 | 6 | 0.401 ± 0.000 | 33.43 ± 4.40 | 0.99 ± 0.00 |
| Wing Weight | 2,000,000 | 10 | 0.072 ± 0.004 | 78.78 ± 9.26 | 1.22 ± 0.00 |

Xeon E5-2680 v3 @ 2.50GHz CPU. We manually inject observational noise into simulated datasets. For query dataset, we constrain the learned noise to be at least 0.1 to regularize the ill-conditioned kernel matrix. For borehole, otl circuit and wing weight datasets, we first learn the hyperparameters and then estimate the RMSE by predicting 40,000 adjacent test points using 60,000 nearby training points due to memory limit.

Table 1 exhibits the prediction accuracy of different GPs measured by RMSE. We find that sgGP consistently outperforms other GPs regardless of dataset size, input dimension and training starting point. Notably, sgGP is able to achieve approximately half the error of SGPR and SVGP on certain datasets like Levy and PM2.5. By comparing to EGP, we conjecture that the implicit regularization effect of sgGP by utilizing correlated minibatches led to smaller RMSE, i.e. the algorithm tends to approach local minimas that have better generalization performance. It is worth noting that 100 epochs of updates are sometimes unnecessary for sgGP as 25 or 50 epochs often result in good performance.

Table 1 also shows the training time of different GPs. The results illustrate the overwhelming training time advantage of sgGP over other GPs, especially when parallel and distributed computing resources are not accessible. As expected, the timing advantage of sgGP scales with dataset size. The time-performance trade-off has been studied in [38] where EGP is shown to have more favorable prediction accuracy than scalable approximation methods at the cost of multiple GPUs with sizable memory on large datasets. Our experiments indicate that sgGP is able to attain preferable hyperparameters to EGP at a much lower computational cost.

Table 2 displays the results of sgGP on toy datasets of size $2 \times 10^6$. Remarkably, it takes around 30 min to train otl circuit dataset using a single CPU core with R functions which are not designed for fast execution. In addition, sgGP enjoys superior training memory efficiency due to its use of minibatches. This experiment justifies that SGD opens up a new data size regime for exploring GPs.

## 8 Conclusion

In this paper, we provide theoretical guarantees for the minibatch SGD for training the Gaussian process (GP) model. In particular, we prove that the parameter iterates converge to the true hyperparameters and a critical point of the full loss function, with rate $O(\frac{1}{K})$ up to a statistical error term depending on minibatch size. Given the correlation structure of GPs, the challenge lies in the bias of stochastic gradient when taking expectation w.r.t. random sampling. Numerical studies support our theoretical results and show that minibatch SGD has better performance than some state-of-the-art methods for various datasets while enjoying huge computational benefits. We finally note that investigating variance reduction techniques in correlated settings might be a promising direction to explore.

## Broader Impact

Practitioners in various areas including, but not limited to, machine learning, statistics and optimization can benefit from applying our proposed framework. Our framework does not use any bias in the data or sensitive information. We do not foresee any negative outcomes on ethical aspects or future societal consequences.

## Acknowledgments and Disclosure of Funding

H. Chen, L. Zheng, and G. Raskutti were partially supported by NSF DMS-1811767. R. Al Kontar was partially supported by NSF CMMI-1931950.

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
