[Supplementary Material]

Supplementary File for

"Stochastic Gradient Descent in Correlated Settings:

A Study on Gaussian Processes"

The supplementary file is organized as follows: Section 1 restates the assumptions and main theorems on the convergence of parameter iterates and the full gradient; Section 2 is devoted to the proofs of the two main theorems, while Section 3 includes the proofs of supporting lemmas; Section 4 includes additional figures from the numerical study.

# 1 Main Theoretical Results

**Assumption 1.1.** *The eigenvalues of kernel function $k$ w.r.t. probability measure $\mathbb{P}$ are $\{Ce^{-bj}\}_{j=0}^{\infty}$, where $b > 0$, and $C \leq 1$ are regarded as constants.*

**Assumption 1.2.** *Both $\boldsymbol{\theta}^*$ and $\boldsymbol{\theta}^{(k)}$ for $0 \leq k \leq K$ lie in $[\theta_{\min}, \theta_{\max}]^2$, where $0 < \theta_{\min} < \theta_{\max}$.*

**Assumption 1.3.** *For all $0 \leq k < K$, $\|g(\boldsymbol{\theta}^{(k)}; \mathbf{X}_{\xi_{k+1}}, \mathbf{y}_{\xi_{k+1}})\|_2 \leq G$ for some $G > 0$.*

**Theorem 1.** *Under Assumptions 1.1 to 1.3, when $m > C$ for some constant $C > 0$, we have the following results under two corresponding conditions on $s_l(m)$:*

1. *If $s_2(m) = m$, $\frac{3}{2\gamma} \leq \alpha_1 \leq \frac{2}{\gamma}$ where $\gamma = \frac{1}{4\theta_{\max}^2}$, then for any $0 < \varepsilon < C\frac{\log \log m}{\log m}$, with probability at least $1 - CK \exp\{-cm^{2\varepsilon}\}$,*

$$(\theta_2^{(K)} - \theta_2^*)^2 \leq \frac{8G^2}{\gamma^2(K+1)} + Cm^{-\frac{1}{2}+\varepsilon}. \tag{1}$$

2. *If in addition to $s_2(m) = m$, $s_1(m)$ is set as $\tau \log m$ where $\tau > \frac{64\theta_{\max}^4}{b\theta_{\min}^4}$, $\frac{3}{2\gamma} \leq \alpha_1 \leq \frac{2}{\gamma}$ where $\gamma$ is defined as in (5), then for any $0 < \varepsilon < \frac{1}{2}$, with probability at least $1 - CK \exp\{-c(\log m)^{2\varepsilon}\}$,*

$$\|\boldsymbol{\theta}^{(K)} - \boldsymbol{\theta}^*\|_2^2 \leq \frac{8G^2}{\gamma^2(K+1)} + C(\log m)^{-\frac{1}{2}+\varepsilon}. \tag{2}$$

*Here $c, C > 0$ depend only on $\theta_{\min}, \theta_{\max}, b$.*

**Theorem 2.** *Under Assumptions 1.1 to 1.3, if $\frac{3}{2\gamma} \leq \alpha_1 \leq \frac{2}{\gamma}$ for $\gamma = \frac{1}{4\theta_{\max}^2}$, $m > C$, $s_2(m) = m$, then for any $0 < \varepsilon < C\frac{\log\log m}{\log m}$, with probability at least $1 - CK\exp\{-cm^{2\varepsilon}\}$,*

$$\|\nabla\ell(\boldsymbol{\theta}^{(K)})\|_2^2 \leq C\left[\frac{G^2}{K+1} + m^{-\frac{1}{2}+\varepsilon}\right], \tag{3}$$

*holds, where $c, C > 0$ depend only on $\theta_{\min}, \theta_{\max}, b$.*

## 2 Proofs of Theorem 1 and Theorem 2

*Proof of Theorem 1.* First we present the following lemma, showing that the loss function has a property similar from strong convexity.

**Lemma 2.1.** *If $s_2(m) = m$, $m > C$ for some $C > 0$, then with probability at least $1 - 2Km^{-c}$, the following claim holds true for $0 \leq k < K$:*

$$\langle \widetilde{\boldsymbol{\theta}}^{(k)} - \widetilde{\boldsymbol{\theta}}^*, \widetilde{g}_k^* \rangle \geq \frac{\gamma}{2}\|\widetilde{\boldsymbol{\theta}}^{(k)} - \widetilde{\boldsymbol{\theta}}^*\|_2^2 - \varepsilon, \tag{4}$$

*where $\widetilde{\boldsymbol{\theta}}^{(k)} = \theta_2^{(k)}$, $\widetilde{\boldsymbol{\theta}}^* = \theta_2^*$, $\widetilde{g}_k^* = (g^*(\boldsymbol{\theta}^{(k)}))_2$, $\gamma = \frac{1}{4\theta_{\max}^2}$, $\varepsilon = \frac{C\log m}{m}$.*

*If in addition to $s_2(m) = m$, we also have $s_1(m) = \tau\log m$ where $\tau > \frac{64\theta_{\max}^4}{b\theta_{\min}^4}$, then with probability at least $1 - 2Km^{-c}$, (4) holds for $\widetilde{\boldsymbol{\theta}}^{(k)} = \boldsymbol{\theta}^{(k)}$, $\widetilde{\boldsymbol{\theta}}^* = \boldsymbol{\theta}^*$, $\widetilde{g}_k^* = g^*(\boldsymbol{\theta})$,*

$$\gamma = \min\left\{\frac{1}{32\tau b\theta_{\max}^2}, \frac{1}{4\theta_{\max}^2} - \frac{2\theta_{\max}^2}{\tau b\theta_{\min}^4}\right\}. \tag{5}$$

*and $\varepsilon = C\frac{\log m}{m}$. Here $C > 0$ depends only on $\theta_{\min}, \theta_{\max}, b$.*

For the first case discussed in Lemma 2.1, define $\widetilde{g}(\boldsymbol{\theta}^{(k)}) = (g(\boldsymbol{\theta}^{(k)}))_2$, and for the second case define $\widetilde{g}(\boldsymbol{\theta}^{(k)}) = g(\boldsymbol{\theta}^{(k)})$. Then let $\widehat{\mathbf{e}}_k = \widetilde{g}(\boldsymbol{\theta}^{(k)}) - \widetilde{g}_k^*$. Due to Lemma 2.1 and Assumption 1.3, we have

$$\begin{aligned}
\|\widetilde{\boldsymbol{\theta}}^{(k)} - \widetilde{\boldsymbol{\theta}}^*\|_2^2 =& \|\widetilde{\boldsymbol{\theta}}^{(k-1)} - \widetilde{\boldsymbol{\theta}}^*\|_2^2 - 2\alpha_k\langle \widetilde{\boldsymbol{\theta}}^{(k-1)} - \widetilde{\boldsymbol{\theta}}^*, \widetilde{g}(\boldsymbol{\theta}^{(k-1)})\rangle + \alpha_k^2\|\widetilde{g}(\boldsymbol{\theta}^{(k-1)})\|_2^2 \\
\leq& \|\widetilde{\boldsymbol{\theta}}^{(k-1)} - \widetilde{\boldsymbol{\theta}}^*\|_2^2(1 - \alpha_k\gamma) + \alpha_k^2 G^2 + 2\alpha_k\left(\varepsilon - \langle \widetilde{\boldsymbol{\theta}}^{(k-1)} - \widetilde{\boldsymbol{\theta}}^*, \widehat{\mathbf{e}}_{k-1}\rangle\right).
\end{aligned} \tag{6}$$

Recall that $\frac{3}{2\gamma} \leq \alpha_1 \leq \frac{2}{\gamma}$, and $\alpha_k = \frac{\alpha_1}{k}$ for all $k \geq 1$. Now we prove the following statement for $k \geq 1$ by induction:

$$\|\widetilde{\boldsymbol{\theta}}^{(k)} - \widetilde{\boldsymbol{\theta}}^*\|_2^2 \leq \frac{2\alpha_1^2 G^2}{k+1} + \sum_{i=0}^{k-1}\eta_{k,i}\left(\varepsilon - \langle\widetilde{\boldsymbol{\theta}}^{(i)} - \widetilde{\boldsymbol{\theta}}^*, \widehat{\mathbf{e}}_i\rangle\right), \tag{7}$$

where $\eta_{k,i} = 2\alpha_{i+1}\prod_{j=i+2}^{k}(1 - \alpha_j\gamma)$. When $k = 1$, by (6) and the fact that $1 - \alpha_1\gamma < 0$,

$$\|\widetilde{\boldsymbol{\theta}}^{(1)} - \widetilde{\boldsymbol{\theta}}^*\|_2^2 \leq \alpha_1^2 G^2 + \eta_{1,0}\left(\varepsilon - \langle\widetilde{\boldsymbol{\theta}}^{(0)} - \widetilde{\boldsymbol{\theta}}^*, \widehat{\mathbf{e}}_0\rangle\right). \tag{8}$$

Assuming (7) holds for $k = l \geq 1$, then due to (6) and the fact that $1 - \alpha_{l+1}\gamma \geq 0$ for $l \geq 1$, we have

$$
\|\widetilde{\boldsymbol{\theta}}^{(l+1)} - \widetilde{\boldsymbol{\theta}}^*\|_2^2
$$

$$
\leq \left( \frac{2\alpha_1^2 G^2}{l+1} + \sum_{i=0}^{l-1} \eta_{l,i} \left( \varepsilon - \langle \widetilde{\boldsymbol{\theta}}^{(i)} - \widetilde{\boldsymbol{\theta}}^*, \widehat{\mathbf{e}}_i \rangle \right) \right) (1 - \alpha_{l+1}\gamma) + \alpha_{l+1}^2 G^2 + 2\alpha_{l+1} \left( \varepsilon - \langle \widetilde{\boldsymbol{\theta}}^{(l)} - \widetilde{\boldsymbol{\theta}}^*, \widehat{\mathbf{e}}_l \rangle \right)
$$

$$
\leq \frac{2\alpha_1^2 G^2 (l+1-\alpha_1\gamma)}{(l+1)^2} + \frac{\alpha_1^2 G^2}{(l+1)^2} + \sum_{i=0}^{l} \eta_{l+1,i} \left( \varepsilon - \langle \widetilde{\boldsymbol{\theta}}^{(i)} - \widetilde{\boldsymbol{\theta}}^*, \widehat{\mathbf{e}}_i \rangle \right)
$$

$$
\leq \frac{2\alpha_1^2 G^2}{l+2} + \sum_{i=0}^{l} \eta_{l+1,i} \left( \varepsilon - \langle \widetilde{\boldsymbol{\theta}}^{(i)} - \widetilde{\boldsymbol{\theta}}^*, \widehat{\mathbf{e}}_i \rangle \right).
$$

$$(9)$$

Here the last two lines are due to range of $\alpha_1$ and the definitions of $\eta_{l,i}$. The next step is to bound $\sum_{i=0}^{K-1} \eta_{K,i} \left( \varepsilon - \langle \widetilde{\boldsymbol{\theta}}^{(i)} - \widetilde{\boldsymbol{\theta}}^*, \widehat{\mathbf{e}}_i \rangle \right)$. First we have

$$
\left| \sum_{i=0}^{K-1} \eta_{K,i} \left( \varepsilon - \langle \widetilde{\boldsymbol{\theta}}^{(i)} - \widetilde{\boldsymbol{\theta}}^*, \widehat{\mathbf{e}}_i \rangle \right) \right|
$$

$$
\leq \frac{2\alpha_1}{K} \sum_{i=0}^{K-1} \|\widetilde{\boldsymbol{\theta}}^{(i)} - \widetilde{\boldsymbol{\theta}}^*\|_2 \|\widehat{\mathbf{e}}_i\|_2 + 2\alpha_1 \varepsilon \qquad (10)
$$

$$
\leq C \left( \max_{0 \leq i \leq K-1} \|\widehat{\mathbf{e}}_i\|_2 + \varepsilon \right).
$$

Note that the distribution of each minibatch $\{\mathbf{X}_{\xi_k+1}, \mathbf{y}_{\xi_k+1}\}_{i=1}^m$ is the same as sampling $m$ independent $\mathbf{x}_i$ from $\mathbb{P}$, and then sampling $\mathbf{y}_{\xi_k+1} \sim \mathcal{N}(0, \mathbf{K}_{\xi_k+1}^*)$, thus we can apply the following lemma on $\widetilde{g}(\boldsymbol{\theta}^{(k)})$ and $\widetilde{g}_k^*$.

**Lemma 2.2.** *[Uniform statistical error]For any $x > 0$, $1 \leq i \leq 2$, we have*

$$
\mathbb{P} \left( \sup_{\boldsymbol{\theta} \in [\theta_{\min}, \theta_{\max}]^2} \frac{n}{s_i(n)} |(\nabla \ell(\boldsymbol{\theta}))_i - (\nabla \ell^*(\boldsymbol{\theta}))_i| > Cx \right) \leq \delta(x). \qquad (11)
$$

*If $s_i(n) = \tau \log n$ for $\tau > \frac{64\theta_{\max}^4}{b\theta_{\min}^4}$, $n > C$ for some $C > 0$, then*

$$
\delta(x) \leq Cn^{-c} + C(\log x)^4 \exp\{-c \log n \min\{x^2, x\}\}.
$$

*If $s_i(n) = n$,*

$$
\delta(x) \leq C(\log x)^4 \exp\{-cn \min\{x^2, x\}\}.
$$

*Here $c, C > 0$ only depend on $\theta_{\min}, \theta_{\max}, b$.*

Therefore, combining Lemma 2.1, Lemma 2.2 and (7) leads to the following conclusion.

1. If $s_2(m) = m$, $m > C$, then for any $0 < \varepsilon < \frac{1}{2}$, with probability at least $1 - CKm^{-c} - CK\exp\{-cm^{2\varepsilon}\}$,

$$(\theta_2^{(K)} - \theta_2^*)^2 \leq \frac{8G^2}{\gamma^2(K+1)} + Cm^{-\frac{1}{2}+\varepsilon}, \tag{12}$$

where $\gamma = \frac{1}{4\theta_{\max}^2}$. Let $\varepsilon < C\frac{\log\log m}{\log m}$, then $K\exp\{-cm^{2\varepsilon}\} \geq CKm^{-c}$, thus the probability term is $1 - CK\exp\{-cm^{2\varepsilon}\}$.

2. If $s_1(m) = \tau\log m$, $s_2(m) = m$, $m > C$, then for any $0 < \varepsilon < \frac{1}{2}$, with probability at least

$$1 - CK\exp\{-c(\log m)^{2\varepsilon}\},$$

we have

$$(\theta_1^{(K)} - \theta_1^*)^2 + (\theta_2^{(K)} - \theta_2^*)^2 \leq \frac{8G^2}{\gamma^2(K+1)} + C(\log m)^{-\frac{1}{2}+\varepsilon}, \tag{13}$$

where $\gamma$ is defined in (5).r bound is $\frac{8G^2}{\gamma^2(K+1)} + C(\log m)^{-\frac{1}{2}+\varepsilon}$.

Here $c, C > 0$ depend only on $\theta_{\min}, \theta_{\max}, b$. $\qquad\qquad\qquad\qquad\qquad\qquad\square$

*Proof of Theorem 2.* We start from bounding $\nabla\ell^*(\boldsymbol{\theta}^{(k)})$, the conditional expectation of $\nabla\ell(\boldsymbol{\theta}^{(k)})$ given $\mathbf{x}_1,\ldots,\mathbf{x}_n$, then control the statistical error $\nabla\ell(\boldsymbol{\theta}^{(k)}) - \nabla\ell^*(\boldsymbol{\theta}^{(k)})$. By the definition of $\nabla\ell^*(\boldsymbol{\theta}^{(k)})$,

$$
\begin{aligned}
\left(\nabla\ell^*(\boldsymbol{\theta}^{(k)})\right)_i &= \frac{1}{2n}\mathrm{tr}\left[\mathbf{K}_n(\boldsymbol{\theta}^{(k)})^{-1}(\mathbf{I}_n - \mathbf{K}_n^*\mathbf{K}_n(\boldsymbol{\theta}^{(k)})^{-1})\frac{\partial\mathbf{K}_n(\boldsymbol{\theta}^{(k)})}{\partial\theta_i^{(k)}}\right] \\
&= \frac{1}{2n}\sum_{l=1}^2(\theta_l^{(k)} - \theta_l^*)\sum_{j=1}^n\frac{\lambda_{lj}\lambda_{ij}}{\left(\sum_{h=1}^2\theta_h^{(k)}\lambda_{hj}\right)^2},
\end{aligned} \tag{14}
$$

where $\lambda_{1j}$ is the $j$th largest eigenvalue of $\mathbf{K}_{f,n}$ and $\lambda_{2j} = 1$. The following lemma provides bounds for $\sum_{j=1}^n\frac{\lambda_{lj}\lambda_{ij}}{\left(\sum_{h=1}^2\theta_h^{(k)}\lambda_{hj}\right)^2}$ for all $1 \leq i, l \leq 2$.

**Lemma 2.3.** *For any $0 < \alpha, \epsilon < 1$, if $n > C(\epsilon)$ for $C > 0$ depending on $b, \epsilon$, then with probability at least $1 - 2n^{-\alpha}$, for any $\boldsymbol{\theta} \in [\theta_{\min}, \theta_{\max}]^2$,*

$$
\begin{aligned}
\frac{\epsilon\log n}{8b\theta_{\max}^2} &\leq \sum_{j=1}^n\frac{\lambda_{1j}^2}{\left(\sum_{h=1}^2\theta_h\lambda_{hj}\right)^2} \leq \frac{2(2+\alpha)}{b\theta_{\min}^2}\log n, \\
\frac{n - C(\alpha)\log n}{4\theta_{\max}^2} &\leq \sum_{j=1}^n\frac{\lambda_{2j}^2}{\left(\sum_{h=1}^2\theta_h\lambda_{hj}\right)^2} \leq \frac{n}{\theta_{\min}^2}, \\
\sum_{j=1}^n\frac{\lambda_{1j}\lambda_{2j}}{\left(\sum_{h=1}^2\theta_h\lambda_{hj}\right)^2} &\leq \frac{5+2\alpha}{7b\theta_{\min}^2}\log n,
\end{aligned} \tag{15}
$$

*where $C(\alpha) > 0$ depends only on $\alpha, b$.*

Apply Lemma 2.3, then for any constant $c > 0$, let $\alpha = c$, (15) holds with probability at least $1 - 2n^{-c}$, if $n > C$ for $C$ depending on $b$. Combining this result and (14) together implies

$$\left|\left(\nabla\ell^*(\boldsymbol{\theta}^{(k)})\right)_1\right| \leq \frac{C\log n}{n} \tag{16}$$

where $C > 0$ depends on $\theta_{\min}, \theta_{\max}, b$. Meanwhile,

$$\left|\left(\nabla\ell^*(\boldsymbol{\theta}^{(k)})\right)_2\right| \leq C\left(|\theta_2^{(k)} - \theta_2^*| + \frac{\log n}{n}\right). \tag{17}$$

Thus we have

$$\|\nabla\ell^*(\boldsymbol{\theta}^{(k)})\|_2^2 \leq C\left[\left(\frac{\log n}{n}\right)^2 + (\theta_2^{(k)} - \theta_2^*)^2\right]. \tag{18}$$

For bounding $\nabla\ell(\boldsymbol{\theta}^{(k)}) - \nabla\ell^*(\boldsymbol{\theta}^{(k)})$, we can apply Lemma 2.2 with $s_i(n) = n$. By (11), Theorem 1 and Lemma 2.2, for any $0 < \varepsilon < C\frac{\log\log m}{\log m}$, if $m > C$, then with probability at least $1 - CK\exp\{-cm^{2\varepsilon}\}$, we have

$$\|\nabla\ell(\boldsymbol{\theta}^{(K)})\|_2^2 \leq C\left[\frac{G^2}{K+1} + m^{-\frac{1}{2}+\varepsilon}\right], \tag{19}$$

where $c, C > 0$ depend only on $\theta_{\min}, \theta_{\max}, b$.

$\square$

# 3    Proofs of Supporting Lemmas

*proof of Lemma 2.1.* Let $\lambda_{1j}^{(k)}$ be the $j$th eigenvalue of $\mathbf{K}_{f,\xi_{k+1}}$, and $\lambda_{2j}^{(k)} = 1$ be the $j$th eigenvalue of $\mathbf{I}_m$, then by the definition of $g^*(\boldsymbol{\theta}^{(k)})$, we have

$$\begin{aligned}
(g^*(\boldsymbol{\theta}^{(k)}))_1 =& \frac{1}{2s_1(m)}\text{tr}\left[\mathbf{K}_{\xi_{k+1}}(\boldsymbol{\theta}^{(k)})^{-1}\left(\mathbf{I}_m - \mathbf{K}_{\xi_{k+1}}(\boldsymbol{\theta}^*)\mathbf{K}_{\xi_{k+1}}(\boldsymbol{\theta}^{(k)})^{-1}\right)\mathbf{K}_{f,\xi_{k+1}}\right]\\
=& \frac{1}{2s_1(m)}\text{tr}\left[\mathbf{K}_{\xi_{k+1}}(\boldsymbol{\theta}^{(k)})^{-1}\left((\theta_1^{(k)} - \theta_1^*)\mathbf{K}_{f,\xi_{k+1}} + (\theta_2^{(k)} - \theta_2^*)\mathbf{I}_m\right)\right.\\
&\left.\mathbf{K}_{\xi_{k+1}}(\boldsymbol{\theta}^{(k)})^{-1}\mathbf{K}_{f,\xi_{k+1}}\right]\\
=& \frac{1}{2s_1(m)}\sum_{l=1}^2(\theta_l^{(k)} - \theta_l^*)\sum_{j=1}^m\frac{\lambda_{lj}^{(k)}\lambda_{1j}^{(k)}}{\left(\sum_{l=1}^2\theta_l^{(k)}\lambda_{lj}^{(k)}\right)^2},
\end{aligned} \tag{20}$$

and

$$\begin{aligned}
(g^*(\boldsymbol{\theta}^{(k)}))_2 =& \frac{1}{2m}\text{tr}\left[\mathbf{K}_{\xi_{k+1}}(\boldsymbol{\theta}^{(k)})^{-1}\left(\mathbf{I}_m - \mathbf{K}_{\xi_{k+1}}(\boldsymbol{\theta}^*)\mathbf{K}_{\xi_{k+1}}(\boldsymbol{\theta}^{(k)})^{-1}\right)\right]\\
=& \frac{1}{2m}\sum_{l=1}^2(\theta_l^{(k)} - \theta_l^*)\sum_{j=1}^m\frac{\lambda_{lj}^{(k)}}{\left(\sum_{l=1}^2\theta_l^{(k)}\lambda_{lj}^{(k)}\right)^2}.
\end{aligned} \tag{21}$$

We prove Lemma 2.1 under two cases separately.

1. $s_1(m) = \tau \log m$, $s_2(m) = m$, $\widetilde{\boldsymbol{\theta}}^{(k)} = \boldsymbol{\theta}^{(k)}$, $\widetilde{\boldsymbol{\theta}}^* = \boldsymbol{\theta}^*$ and $\widetilde{g}_k^* = g^*(\boldsymbol{\theta}^{(k)})$

   Under this case, we can write $\langle \widetilde{\boldsymbol{\theta}}^{(k)} - \widetilde{\boldsymbol{\theta}}^*, \widetilde{g}_k^* \rangle$ as

   $$\langle \widetilde{\boldsymbol{\theta}}^{(k)} - \widetilde{\boldsymbol{\theta}}^*, \widetilde{g}_k^* \rangle = (\widetilde{\boldsymbol{\theta}}^{(k)} - \widetilde{\boldsymbol{\theta}}^*)^\top \mathbf{A} (\widetilde{\boldsymbol{\theta}}^{(k)} - \widetilde{\boldsymbol{\theta}}^*),$$

   where each entry $A_{ij}$ of $\mathbf{A} \in \mathbb{R}^{2 \times 2}$ is defined as follows:

   $$A_{11} = \frac{1}{2\tau \log m} \sum_{j=1}^{m} \frac{\lambda_{1j}^{(k)2}}{\left( \sum_{l=1}^{2} \theta_l^{(k)} \lambda_{lj}^{(k)} \right)^2},$$

   $$A_{12} = \frac{1}{2\tau \log m} \sum_{j=1}^{m} \frac{\lambda_{1j}^{(k)}}{\left( \sum_{l=1}^{2} \theta_l^{(k)} \lambda_{lj}^{(k)} \right)^2},$$

   $$A_{21} = \frac{1}{2m} \sum_{j=1}^{m} \frac{\lambda_{1j}^{(k)}}{\left( \sum_{l=1}^{2} \theta_l^{(k)} \lambda_{lj}^{(k)} \right)^2},$$

   $$A_{22} = \frac{1}{2m} \sum_{j=1}^{m} \frac{1}{\left( \sum_{l=1}^{2} \theta_l^{(k)} \lambda_{lj}^{(k)} \right)^2}.$$

Note that the distribution of each minibatch $\mathbf{X}_{\xi_{k+1}}$ can be seen as $m$ independent samples from $\mathbb{P}$, thus we can still apply Lemma 2.3, but substituting $n$ by $m$. Apply (15) in Lemma 2.3 with $\epsilon = \frac{1}{2}$, then for any $0 < \alpha < 1$, with probability at least $1 - 2m^{-\alpha}$, we have

$$A_{11} \geq \frac{1}{32\tau b \theta_{\max}^2}, \quad A_{22} \geq \frac{1}{8\theta_{\max}^2} \left( 1 - \frac{C \log m}{m} \right),$$
$$A_{12} \leq \frac{1}{2\tau b \theta_{\min}^2}, \quad A_{21} \leq \frac{\log m}{2b\theta_{\min}^2 m}. \tag{22}$$

Also note that for any $\omega > 0$,

$$(\widetilde{\boldsymbol{\theta}}^{(k)} - \widetilde{\boldsymbol{\theta}}^*)^\top \mathbf{A} (\widetilde{\boldsymbol{\theta}}^{(k)} - \widetilde{\boldsymbol{\theta}}^*)$$
$$\geq \left( A_{11} - \frac{(A_{12} + A_{21})\omega}{2} \right) (\theta_1^{(k)} - \theta_1^*)^2 + \left( A_{22} - \frac{(A_{12} + A_{21})}{2\omega} \right) (\theta_2^{(k)} - \theta_2^*)^2 \tag{23}$$

Let $\omega = \frac{\theta_{\min}^2}{16\theta_{\max}^2}$, then by (22) and (23), one can show that

$$\langle \widetilde{\boldsymbol{\theta}}^{(k)} - \widetilde{\boldsymbol{\theta}}^*, \widetilde{g}_k^* \rangle$$
$$\geq \frac{1}{64\tau b \theta_{\max}^2} (\theta_1^{(k)} - \theta_1^*)^2 + \left( \frac{1}{8\theta_{\max}^2} - \frac{4\theta_{\max}^2}{\tau b \theta_{\min}^4} \right) (\theta_2^{(k)} - \theta_2^*)^2 - C\frac{\log m}{m} \tag{24}$$
$$\geq \frac{\gamma}{2} \|\widetilde{\boldsymbol{\theta}}^{(k)} - \widetilde{\boldsymbol{\theta}}^*\|_2^2 - C\frac{\log m}{m},$$

where
$$\gamma = \min\left\{\frac{1}{32\tau b \theta_{\max}^2}, \frac{1}{4\theta_{\max}^2} - \frac{2\theta_{\max}^2}{\tau b \theta_{\min}^4}\right\}, \tag{25}$$
and $C > 0$ depends on $\theta_{\min}, \theta_{\max}, b$. It is guaranteed that $\gamma > 0$ Since we have assumed
$$\tau > \frac{64\theta_{\max}^4}{b\theta_{\min}^4}.$$

Therefore, if $m > C$, for any $0 < \alpha < 1$, with probability $1 - 2m^{-\alpha}$, the following claims holds true:
$$\langle \widetilde{\boldsymbol{\theta}}^{(k)} - \widetilde{\boldsymbol{\theta}}^*, \widetilde{g}_k^* \rangle \geq \frac{\gamma}{2}\|\widetilde{\boldsymbol{\theta}}^{(k)} - \widetilde{\boldsymbol{\theta}}^*\|_2^2 - \varepsilon, \tag{26}$$
where $\varepsilon = C\frac{\log m}{m}$ for some constant $C > 0$ depending on $\theta_{\min}, \theta_{\max}, b$.

2. $s_2(m) = m$, $\widetilde{\boldsymbol{\theta}}^{(k)} = \theta_2^{(k)}$, $\widetilde{\boldsymbol{\theta}}^* = \theta_2^*$ and $\widetilde{g}_k^* = (g^*(\boldsymbol{\theta}^{(k)}))_2$

   Under this case, we can still apply (15) in Lemma 2.3. Following similar arguments from the first case, one can show that with probability at least $1 - 2m^{-c}$,
$$\langle \widetilde{\boldsymbol{\theta}}^{(k)} - \widetilde{\boldsymbol{\theta}}^*, \widetilde{g}_k^* \rangle \geq \frac{\gamma}{2}(\theta_2^{(k)} - \theta_2^*)^2 - \varepsilon, \tag{27}$$
   where $\gamma = \frac{1}{4\theta_{\max}^2}$, $\varepsilon = \frac{C\log m}{m}$, if $m > C$. Here $C > 0$ depends only on $\theta_{\min}, \theta_{\max}, b$.

$\square$

*proof of Lemma 2.2.* Without loss of generality, we start from bounding $(\nabla\ell(\boldsymbol{\theta}))_i - (\nabla\ell^*(\boldsymbol{\theta}))_i$ for an arbitrary $1 \leq i \leq 2$. Let $K_{f,n}^{(1)} = K_{f,n}$ and $K_{f,n}^{(2)} = I_n$. By the definition of $\nabla\ell(\boldsymbol{\theta})$ and $\nabla\ell^*(\boldsymbol{\theta})$, we have

$$
\begin{aligned}
&(\nabla\ell(\boldsymbol{\theta}))_i - (\nabla\ell^*(\boldsymbol{\theta}))_i \\
&= -\frac{1}{2n}\left[\mathbf{y}_n^\top \mathbf{K}_n^{-1}(\boldsymbol{\theta})\mathbf{K}_{f,n}^{(i)}\mathbf{K}_n^{-1}(\boldsymbol{\theta})\mathbf{y}_n - \text{tr}\left(\mathbf{K}_n(\boldsymbol{\theta})^{-1}\mathbf{K}_{f,n}^{(i)}\mathbf{K}_n(\boldsymbol{\theta})^{-1}\mathbf{K}_n^*\right)\right] \\
&= -\left((\mathbf{K}_n^*)^{-\frac{1}{2}}\mathbf{y}_n\right)^\top \mathbf{A}(\boldsymbol{\theta})\left((\mathbf{K}_n^*)^{-\frac{1}{2}}\mathbf{y}_n\right) + \text{tr}(\mathbf{A}(\boldsymbol{\theta})),
\end{aligned} \tag{28}
$$

where $\mathbf{A}(\boldsymbol{\theta}) = \frac{1}{2n}\mathbf{K}_n^{*\frac{1}{2}}\mathbf{K}_n^{-1}(\boldsymbol{\theta})\mathbf{K}_{f,n}^{(i)}\mathbf{K}_n^{-1}(\boldsymbol{\theta})\mathbf{K}_n^{*\frac{1}{2}}$. Since $K_{f,n}^{(j)}$ can be simultaneously diagonalized, we can write $\mathbf{K}_{f,n}^{(j)} = \mathbf{P}^\top \boldsymbol{\Lambda}_j \mathbf{P}$ for all $j$, where $\mathbf{P}$ is an orthogonal matrix and $\boldsymbol{\Lambda}_j$ is a diagonal matrix consisting of the eigenvalues of $\mathbf{K}_{f,n}^{(j)}$. Then we have

$$\mathbf{A}(\boldsymbol{\theta}^{(k)}) = \mathbf{P}^\top \frac{1}{2n}\left(\sum_{l=1}^{2}\theta_l^*\boldsymbol{\Lambda}_l\right)\left(\sum_{l=1}^{2}\theta_l\boldsymbol{\Lambda}_l\right)^{-2}\boldsymbol{\Lambda}_i\mathbf{P}. \tag{29}$$

Let $\mathbf{z}_n = \mathbf{P}(\mathbf{K}_n^*)^{-\frac{1}{2}}\mathbf{y}_n$, and $\boldsymbol{\Lambda}(\boldsymbol{\theta}) = \left(\sum_{l=1}^{2}\theta_l^*\boldsymbol{\Lambda}_l\right)\left(\sum_{l=1}^{2}\theta_l\boldsymbol{\Lambda}_l\right)^{-2}\boldsymbol{\Lambda}_i$, where $\theta_l$ is the $l$th entry of $\boldsymbol{\theta}$, then our goal is to derive a bound for

$$\sup_{\boldsymbol{\theta}\in[\theta_{\min},\theta_{\max}]^2}\frac{1}{2n}\left|\mathbf{z}_n^\top\boldsymbol{\Lambda}(\boldsymbol{\theta})\mathbf{z}_n - \text{tr}(\boldsymbol{\Lambda}(\boldsymbol{\theta}))\right|.$$

We claim that there exists an $\varepsilon$-net $\{\boldsymbol{\theta}_\varepsilon^{(1)}, \ldots, \boldsymbol{\theta}_\varepsilon^{(N)}\}$ of $[\theta_{\min}, \theta_{\max}]^2$ under $\|\cdot\|_\infty$, with size $N = (1 + \frac{(\theta_{\max} - \theta_{\min})}{\varepsilon})^2$. That is to say, for any $\boldsymbol{\theta} \in [\theta_{\min}, \theta_{\max}]^2$, $\exists \widetilde{\boldsymbol{\theta}} \in \{\boldsymbol{\theta}_\varepsilon^{(1)}, \ldots, \boldsymbol{\theta}_\varepsilon^{(N)}\}$ such that $\boldsymbol{\Delta} = \boldsymbol{\theta} - \widetilde{\boldsymbol{\theta}}$ satisfies $\|\boldsymbol{\Delta}\|_\infty \leq \varepsilon$. The following proof of this claim is very similar to the proof of Lemma 5.2 in [2].

Define $\boldsymbol{\theta}_c = (\frac{\theta_{\min} + \theta_{\max}}{2}, \ldots, \frac{\theta_{\min} + \theta_{\max}}{2}) \in \mathbb{R}^2$, then an alternative way to represent $[\theta_{\min}, \theta_{\max}]^2$ is $\boldsymbol{\theta}_c + \mathbb{B}_\infty(\frac{\theta_{\max} - \theta_{\min}}{2})$. Let $\{\boldsymbol{\theta}_\varepsilon^{(1)}, \ldots, \boldsymbol{\theta}_\varepsilon^{(N)}\}$ be a maximal $\varepsilon$-separated subset of $\boldsymbol{\theta}_c + \mathbb{B}_\infty(\frac{\theta_{\max} - \theta_{\min}}{2})$ (not the iterates of the SGD algorithm), which means that it is an $\varepsilon$-net of $\boldsymbol{\theta}_c + B_\infty(\frac{\theta_{\max} - \theta_{\min}}{2})$, and $\forall 1 \leq i \neq j \leq N$, $\|\boldsymbol{\theta}_\varepsilon^{(i)} - \boldsymbol{\theta}_\varepsilon^{(j)}\|_\infty \geq \varepsilon$. Consider the $\ell_\infty$ balls with centers $\{\boldsymbol{\theta}_\varepsilon^{(i)}\}_{i=1}^N$ and radius $\frac{\varepsilon}{2}$, then these balls are disjoint and are subsets of $\boldsymbol{\theta}_c + \mathbb{B}_\infty(\frac{\theta_{\max} - \theta_{\min} + \varepsilon}{2})$. Thus the sum of volumes of these balls is bounded by that of $\boldsymbol{\theta}_c + \mathbb{B}_\infty(\frac{\theta_{\max} - \theta_{\min} + \varepsilon}{2})$, which finishes the proof of

$$N \leq \left(1 + \frac{\theta_{\max} - \theta_{\min}}{\varepsilon}\right)^2, \tag{30}$$

In the following we linearize $\boldsymbol{\Lambda}(\boldsymbol{\theta}) = \boldsymbol{\Lambda}(\widetilde{\boldsymbol{\theta}} + \boldsymbol{\Delta})$ based on the Taylor series expression of each of its diagonal entries, so that the upper bound for $\left|\mathbf{z}_n^\top \boldsymbol{\Lambda}(\boldsymbol{\theta}) \mathbf{z}_n - \mathrm{tr}(\boldsymbol{\Lambda}(\boldsymbol{\theta}))\right|$ can be implied by some bounds related to $\widetilde{\boldsymbol{\theta}}$. For any $1 \leq j \leq m$, denote the $j$th diagonal entry of $\boldsymbol{\Lambda}_l$ by $\lambda_{lj}$ which is independent of $\boldsymbol{\theta}$, then the $j$th diagonal entry of $\boldsymbol{\Lambda}(\boldsymbol{\theta})$ can be written as follows:

$$\boldsymbol{\Lambda}_{jj}(\boldsymbol{\theta}) = \frac{\sum_{l=1}^2 \lambda_{lj} \lambda_{ij} \theta_l^*}{\left(\sum_{l=1}^2 \lambda_{lj} \theta_l\right)^2}. \tag{31}$$

Meanwhile, let $\Delta_l$ and $\widetilde{\theta}_l$ be the $l$th entry of $\boldsymbol{\Delta}$ and $\boldsymbol{\theta}$, then one can show that

$$
\begin{aligned}
\frac{1}{\left(\sum_{l=1}^2 \lambda_{lj} \theta_l\right)^2} &= \frac{1}{\left(\sum_{l=1}^2 \lambda_{lj} \widetilde{\theta}_l + \sum_{l=1}^2 \lambda_{lj} \Delta_l\right)^2} \\
&= \left(\sum_{l=1}^2 \lambda_{lj} \widetilde{\theta}_l\right)^{-2} \left(1 + \frac{\sum_{l=1}^2 \lambda_{lj} \Delta_l}{\sum_{l=1}^2 \lambda_{lj} \widetilde{\theta}_l}\right)^{-2} \\
&= \left(\sum_{l=1}^2 \lambda_{lj} \widetilde{\theta}_l\right)^{-2} \sum_{h=0}^{H-1} \frac{h+1}{\left(-\sum_{l=1}^2 \lambda_{lj} \widetilde{\theta}_l\right)^h} \left(\sum_{l=1}^2 \lambda_{lj} \Delta_l\right)^h \\
&\quad + \frac{H+1}{(1+\xi)^{H+2} \left(-\sum_{l=1}^2 \lambda_{lj} \widetilde{\theta}_l\right)^{H+2}} \left(\sum_{l=1}^2 \lambda_{lj} \Delta_l\right)^H \\
&= \left(\sum_{l=1}^2 \lambda_{lj} \widetilde{\theta}_l\right)^{-2} \left(\sum_{h_1 + h_2 \leq H-1} \alpha_{h_1, h_2}^{(j)} \prod_{l=1}^2 \Delta_l^{h_l} + \mathrm{RES}_H^{(j)}(\boldsymbol{\theta})\right),
\end{aligned} \tag{32}
$$

where the third equality holds if $\left|\sum_{l=1}^2 \lambda_{lj} \Delta_l\right| < \sum_{l=1}^2 \lambda_{lj} \widetilde{\theta}_l$, which is implied by $\|\boldsymbol{\Delta}\|_\infty \leq \theta_{\min}$,

and we will choose $\varepsilon$ small enough to satisfy this. Here $\xi$ lies between 0 and $\frac{\sum_{l=1}^2 \lambda_{lj} \Delta_l}{\sum_{l=1}^2 \lambda_{lj} \widetilde{\theta_l}}$,

$$\alpha_{h_1,h_2}^{(j)} = \frac{(\sum_{l=1}^2 h_l + 1)! \prod_{l=1}^2 \lambda_{lj}^{h_l}}{h_1! h_2! (-\sum_{l=1}^2 \lambda_{lj} \widetilde{\theta_l})^{\sum_{l=1}^2 h_l}},$$

$$\mathrm{RES}_H^{(j)}(\boldsymbol{\theta}) = \sum_{h_1+h_2=H} \frac{(H+1)! \prod_{l=1}^2 \lambda_{lj}^{h_l} \Delta_l^{h_l}}{h_1! h_2! (1+\xi)^{H+2} (-\sum_{l=1}^2 \lambda_{lj} \widetilde{\theta_l})^H}. \qquad (33)$$

The quantities above satisfy

$$|\alpha_{h_1,h_2}^{(j)}| \leq \left( \sum_{l=1}^2 h_l + 1 \right) \left( \frac{2}{\theta_{\min}} \right)^{\sum_{l=1}^2 h_l}, |\mathrm{RES}_H^{(j)}(\boldsymbol{\theta})| \leq (H+1) \left( \frac{2\varepsilon}{\theta_{\min}} \right)^H, \qquad (34)$$

since $\sum_{h_1+h_2=h} \frac{h!}{h_1! h_2!} = 2^h$. Define the following diagonal matrices: $\boldsymbol{\Lambda}^{(h_1,h_2)}(\widetilde{\boldsymbol{\theta}}), \boldsymbol{\Lambda}^{(H)}(\boldsymbol{\theta}) \in \mathbb{R}^{n \times n}$ are with diagonal entries

$$\boldsymbol{\Lambda}_{jj}^{(h_1,h_2)}(\widetilde{\boldsymbol{\theta}}) = \alpha_{h_1,h_2}^{(j)} \boldsymbol{\Lambda}_{jj}(\widetilde{\boldsymbol{\theta}}), \boldsymbol{\Lambda}_{jj}^{(H)}(\boldsymbol{\theta}) = \mathrm{RES}_H^{(j)}(\boldsymbol{\theta}) \boldsymbol{\Lambda}_{jj}(\widetilde{\boldsymbol{\theta}}). \qquad (35)$$

Then we can write

$$\boldsymbol{\Lambda}(\boldsymbol{\theta}) = \sum_{h_1+h_2 \leq H-1} \boldsymbol{\Lambda}^{(h_1,h_2)}(\widetilde{\boldsymbol{\theta}}) \prod_{l=1}^2 \Delta_l^{h_l} + \boldsymbol{\Lambda}^{(H)}(\boldsymbol{\theta}),$$

and thus

$$\left| \mathbf{z}_n^\top \boldsymbol{\Lambda}(\boldsymbol{\theta}) \mathbf{z}_n - \mathrm{tr}(\boldsymbol{\Lambda}(\boldsymbol{\theta})) \right|$$
$$\leq \max_{1 \leq k \leq N} \sum_{h_1+h_2 \leq H-1} \varepsilon^{\sum_{l=1}^2 h_l} \left| \mathbf{z}_n^\top \boldsymbol{\Lambda}^{(h_1,h_2)}(\boldsymbol{\theta}_\varepsilon^{(k)}) \mathbf{z}_n - \mathrm{tr}(\boldsymbol{\Lambda}^{(h_1,h_2)}(\boldsymbol{\theta}_\varepsilon^{(k)})) \right| \qquad (36)$$
$$+ \left| \mathbf{z}_n^\top \boldsymbol{\Lambda}^{(H)}(\boldsymbol{\theta}) \mathbf{z}_n - \mathrm{tr}(\boldsymbol{\Lambda}^{(H)}(\boldsymbol{\theta})) \right|.$$

In order to provide an upper bound for the first term above, we first bound

$$\left| \mathbf{z}_n^\top \boldsymbol{\Lambda}^{(h_1,h_2)}(\boldsymbol{\theta}_\varepsilon^{(k)}) \mathbf{z}_n - \mathrm{tr}(\boldsymbol{\Lambda}^{(h_1,h_2)}(\boldsymbol{\theta}_\varepsilon^{(k)})) \right|$$

for an arbitrary $k$. First note that for any $1 \leq k \leq N$,

$$\|\boldsymbol{\Lambda}(\boldsymbol{\theta}_\varepsilon^{(k)})\|_2 = \max_j \frac{\sum_{l=1}^2 \lambda_{lj} \lambda_{ij} \theta_l^*}{\left( \sum_{l=1}^2 \lambda_{lj} \theta_l^{(k)} \right)^2} \leq \frac{\theta_{\max}}{\theta_{\min}^2}. \qquad (37)$$

While for $\|\boldsymbol{\Lambda}(\boldsymbol{\theta}_\varepsilon^{(k)})\|_F^2$, one can show that

$$\|\boldsymbol{\Lambda}(\boldsymbol{\theta}_\varepsilon^{(k)})\|_F^2 \leq \theta_{\max}^2 \sum_{j=1}^n \frac{(\sum_{l=1}^2 \lambda_{lj} \lambda_{ij})^2}{(\sum_{l=1}^2 \lambda_{lj} \theta_l^{(k)})^4}$$

$$\leq C \sum_{j=1}^n \frac{\sum_{l=1}^2 \lambda_{lj} \lambda_{ij}}{(\sum_{l=1}^2 \lambda_{lj} \theta_l)^2}. \qquad (38)$$

Let

$$t_i(n) = \sum_{j=1}^{n} \frac{\sum_{l=1}^{2} \lambda_{lj} \lambda_{ij}}{(\sum_{l=1}^{2} \lambda_{lj} \theta_l)^2},$$

then a deterministic bound for $t_i(n)$ is

$$t_i(n) \le Cn, \tag{39}$$

while applying Lemma 2.3 leads to

$$t_1(n) \le C \log n, \text{ and } t_2(n) \le Cn, \tag{40}$$

with probability at least $1 - 2n^{-c}$ for any constant $c > 0$, if $n > C$. Here $C > 0$ depends only on $b, \theta_{\min}, \theta_{\max}$. Therefore, by the definition of $\mathbf{\Lambda}^{(h_1,h_2)}(\boldsymbol{\theta}_\varepsilon^{(k)})$, for any $\boldsymbol{\theta}_\varepsilon^{(k)}$,

$$
\begin{aligned}
\|\mathbf{\Lambda}^{(h_1,h_2)}(\boldsymbol{\theta}_\varepsilon^{(k)})\|_2 &\le C \left( \sum_{l=1}^{2} h_l + 1 \right) \left( \frac{2}{\theta_{\min}} \right)^{\sum_{l=1}^{2} h_l}, \\
\|\mathbf{\Lambda}^{(h_1,h_2)}(\boldsymbol{\theta}_\varepsilon^{(k)})\|_F^2 &\le C \left( \sum_{l=1}^{2} h_l + 1 \right)^2 \left( \frac{2}{\theta_{\min}} \right)^{2 \sum_{l=1}^{2} h_l} t_i(n).
\end{aligned}
\tag{41}
$$

Let $\varepsilon = \frac{\theta_{\min}}{2eH}$, then by applying Hanson-wright's inequality, one can show that with probability at least $1 - 2 \exp\{-c \min\{t, \frac{t^2}{t_i(n)}\}\}$,

$$\left| \mathbf{z}_n^{\top} \mathbf{\Lambda}^{(h_1,...,h_2)}(\boldsymbol{\theta}_\varepsilon^{(k)}) \mathbf{z}_n - \mathrm{tr}(\mathbf{\Lambda}^{(h_1,...,h_2)}(\boldsymbol{\theta}_\varepsilon^{(k)})) \right| \le (e\varepsilon)^{-\sum_{l=1}^{2} h_l} t, \tag{42}$$

where $c > 0$ depends on $\theta_{\min}, \theta_{\max}, b$. Meanwhile, the following lemma provides an upper bound for the residual term:

**Lemma 3.1.** $\exists\, c, C > 0$ *depending only on* $\theta_{\min}, \theta_{\max}, b$ *such that,*

$$
\begin{aligned}
&\mathbb{P}\left[ \sup_{\boldsymbol{\theta} \in [\theta_{\min}, \theta_{\max}]^2} |\mathbf{z}_n^{\top} \mathbf{\Lambda}^{(H)}(\boldsymbol{\theta}) \mathbf{z}_n - \mathrm{tr}(\mathbf{\Lambda}^{(H)}(\boldsymbol{\theta}))| > e^{-H}(t + Ct_i(n)) \right] \\
&\le 2 \exp\left\{ -c \min\left\{ \frac{t^2}{t_i(n)}, t \right\} \right\}.
\end{aligned}
\tag{43}
$$

Now we take a union bound for each term in (36), then with probability at least

$$
\begin{aligned}
&1 - 2 \left( \binom{H+2}{2} N + 1 \right) \exp\left\{ -c \min\left\{ \frac{t^2}{t_i(n)}, t \right\} \right\} \\
&\ge 1 - CH^4 \exp\left\{ -c \min\left\{ \frac{t^2}{t_i(n)}, t \right\} \right\},
\end{aligned}
\tag{44}
$$

we have

$$
\begin{aligned}
&\sup_{\boldsymbol{\theta} \in [\theta_{\min}, \theta_{\max}]^2} \left| \mathbf{z}_n^{\top} \mathbf{\Lambda}(\boldsymbol{\theta}) \mathbf{z}_n - \mathrm{tr}(\mathbf{\Lambda}(\boldsymbol{\theta})) \right| \\
&\le t \left[ \sum_{h=0}^{H-1} \binom{h+2}{2} e^{-h} + e^{-H} \right] + Ce^{-H} t_i(n) \\
&\le C(t + e^{-H} t_i(n)).
\end{aligned}
\tag{45}
$$

If $s_i(n) = \tau \log n$ for some $\tau > \frac{64\theta_{\max}^4}{b\theta_{\min}^4}$, we apply the probabilistic bound (40) on $t_i(n)$. For any $x > 0$, let $H = \log \frac{1}{x}$ and $t = x \log n$, then with probability at least

$$1 - Cn^{-c} - C(\log x)^4 \exp\left\{-c \log n \min\left\{x^2, x\right\}\right\},$$

we have

$$\sup_{\boldsymbol{\theta} \in [\theta_{\min}, \theta_{\max}]^2} \left|\mathbf{z}_n^\top \boldsymbol{\Lambda}(\boldsymbol{\theta})\mathbf{z}_n - \mathrm{tr}(\boldsymbol{\Lambda}(\boldsymbol{\theta}))\right| \leq Cx \log n, \tag{46}$$

which implies

$$\sup_{\boldsymbol{\theta} \in [\theta_{\min}, \theta_{\max}]^2} \frac{n}{s_i(n)} \left|(\nabla\ell(\boldsymbol{\theta}))_i - (\nabla\ell^*(\boldsymbol{\theta}))_i\right| \leq Cx. \tag{47}$$

Otherwise, if $s_i(n) = n$, we apply the deterministic bound (39) on $t_i(n)$. For any $x > 0$, let $H = \log\frac{1}{x}$ and $t = xn$, then with probability at least

$$1 - C(\log x)^4 \exp\left\{-cn \min\left\{x^2, x\right\}\right\},$$

we have

$$\sup_{\boldsymbol{\theta} \in [\theta_{\min}, \theta_{\max}]^2} \left|\mathbf{z}_n^\top \boldsymbol{\Lambda}(\boldsymbol{\theta})\mathbf{z}_n - \mathrm{tr}(\boldsymbol{\Lambda}(\boldsymbol{\theta}))\right| \leq Cxn, \tag{48}$$

which implies

$$\frac{n}{s_i(n)} \left|(\nabla\ell(\boldsymbol{\theta}))_i - (\nabla\ell^*(\boldsymbol{\theta}))_i\right| \leq Cx. \tag{49}$$

$\square$

*proof of Lemma 2.3.* In order to prove Lemma 2.3, we need to derive upper and lower bounds for $\lambda_{1j}$ w.h.p. First we restate Theorem 1 and Theorem 4 in [1] on the bounds for $\lambda_{1j}$ in the following:

**Lemma 3.2.** *Let $k$ be a Mercer kernel on a probability space $\mathcal{X}$ with probability measure $\mathbb{P}$, satisfying $k(x, x) \leq 1$ for all $x \in \mathcal{X}$, with eigenvalues $\{\lambda_i^*\}_{i=1}^{\infty}$. Let $\mathbf{K}_{f,n} \in \mathbb{R}^{n \times n}$ be the empirical kernel matrix evaluated on data $\{\mathbf{x}_1, \ldots, \mathbf{x}_n\}$ i.i.d. sampled from $\mathbb{P}$, then the eigenvalues $\lambda_i(\mathbf{K}_{f,n})$ satisfies the following bound for $1 \leq j, r \leq n$:*

$$\left|\frac{\lambda_j(\mathbf{K}_{f,n})}{n} - \lambda_j^*\right| \leq \lambda_j^* C(r, n) + E(r, n),$$

*and for any $1 \leq r \leq n$, there are two bounds for $C(r, n), E(r, n)$:*

*(i) With probability at least $1 - \delta$,*

$$C(r, n) < r\sqrt{\frac{2}{n\lambda_r^*} \log \frac{2r(r+1)}{\delta}} + \frac{4r}{3n\lambda_r^*} \log \frac{2r(r+1)}{\delta},$$

$$E(r, n) < \lambda_r^* + \sum_{i=r+1}^{\infty} \lambda_i^* + \sqrt{\frac{2\sum_{i=r+1}^{\infty}\lambda_i^*}{n} \log \frac{2}{\delta}} + \frac{2}{3n} \log \frac{2}{\delta}; \tag{50}$$

*(ii) With probability at least $1 - \delta$,*

$$C(r,n) < r\sqrt{\frac{r(r+1)}{n\delta\lambda_r^*}}, \quad E(r,n) < \lambda_r^* + \sum_{i=r+1}^{\infty} \lambda_i^* + \sqrt{\frac{2\sum_{i=r+1}^{\infty}\lambda_i^*}{n\delta}}. \tag{51}$$

We consider the following upper bound for $\lambda_{lj}$ that could be useful in later arguments. First we apply Lemma 3.2 on $\mathbf{K}_{f,n}$. In particular, plug $r = j$ for each $1 \leq j \leq n$ into (51) and let $\delta = n^{-(1+\alpha)}$ for some $\alpha > 0$. Then with probability at least $1 - n^{-\alpha}$, for all $1 \leq j \leq n$,

$$1 + C(j,n) < C^{-\frac{1}{2}}j^2 n^{\frac{\alpha}{2}}e^{\frac{bj}{2}}\sqrt{\frac{j+1}{j}} + 1 < Cj^2 n^{\frac{\alpha}{2}}e^{\frac{bj}{2}},$$

$$E(r,n) < \frac{Ce^{-bj}}{1 - e^{-b}} + \sqrt{\frac{2Ce^{-b}}{1 - e^{-b}}}e^{-\frac{bj}{2}}n^{\frac{\alpha}{2}}.$$

Thus we have

$$\lambda_{lj} \leq \left(Cj^2 + \frac{Cn^{-\frac{\alpha}{2}}e^{-\frac{b}{2}j}}{1 - e^{-b}} + \sqrt{\frac{2Ce^{-b}}{1 - e^{-b}}}\right)n^{1+\frac{\alpha}{2}}e^{-\frac{bj}{2}} \tag{52}$$

$$\leq C(\eta)n^{1+\frac{\alpha}{2}}e^{-\frac{bj}{2\eta}},$$

where the last line holds for any $\eta > 1$, and $C(\eta) > 0$ depends on $b, \eta$. We will specify $\eta$ later to suit our needs.

While for lower bounding $\lambda_{1j}$, we apply (50) in Lemma 3.2 with $r = \frac{\epsilon}{b}\log n$ for some $0 < \epsilon < 1$, and $\delta = n^{-\alpha}$ for some $0 < \alpha < 1$. Then when $n > C(\epsilon)$ for some constant $C(\epsilon) > 0$ depending on $b, \epsilon$, with probability at least $1 - n^{-\alpha}$,

$$C(r,n) < r\sqrt{\frac{2\log[2r(r+1)n^{\alpha}]}{Cn^{1-\epsilon}}} + \frac{4r\log[2r(r+1)n^{\alpha}]}{3Cn^{1-\epsilon}} < \frac{1}{2},$$

$$E(r,n) < \frac{C}{(1 - e^{-b})n^{\epsilon}} + \sqrt{\frac{2Ce^{-b}}{1 - e^{-b}}}\sqrt{\frac{\log 2n^{\alpha}}{n^{1+\epsilon}}} + \frac{2}{3n}\log 2n^{\alpha} < Cn^{-\epsilon},$$

thus $\lambda_{1j} \geq \frac{C}{2}ne^{-bj} - Cn^{1-\epsilon}$ for $C > 0$ depending on $b$.

Therefore, for any $0 < \epsilon, \alpha < 1$, if $n > C(\epsilon)$ for $C(\epsilon) > 0$ depending on $b, \epsilon$, then with probability at least $1 - n^{-\alpha}$,

$$\lambda_{1j} \geq \frac{C}{2}ne^{-bj} - Cn^{1-\epsilon}, \tag{53}$$

holds for $1 \leq j \leq n$, where $C > 0$ depends on $b$. Now we are ready to prove the bounds for $\sum_{j=1}^{n}\frac{\lambda_{lj}\lambda_{l'j}}{\left(\sum_{h=1}^{2}\theta_h\lambda_{hj}\right)^2}$ for $1 \leq l, l' \leq 2$.

1. $l = l' = 1$

First we derive an upper bound. Let $\eta = \frac{3}{2}$ in (52), then we have

$$\sum_{j=1}^{n} \frac{\lambda_{1j}^2}{\left(\sum_{h=1}^{2} \theta_h \lambda_{hj}\right)^2} \leq \frac{|\{n^{1+\frac{\alpha}{2}} e^{-\frac{bj}{3}} > 1\}|}{\theta_{\min}^2} + \sum_{n^{1+\frac{\alpha}{2}} e^{-\frac{bj}{3}} \leq 1} \frac{\lambda_{1j}^2}{\theta_{\min}^2}$$

$$\leq \frac{|\{n^{1+\frac{\alpha}{2}} e^{-\frac{bj}{3}} > 1\}|}{\theta_{\min}^2} + \frac{Cn^{2+\alpha}}{\theta_{\min}^2} \sum_{e^{-\frac{2bj}{3}} \leq n^{-2-\alpha}} e^{-\frac{2b}{3}j}. \tag{54}$$

Since

$$n^{1+\frac{\alpha}{2}} e^{-\frac{bj}{3}} > 1 \Rightarrow j < \frac{6+3\alpha}{2b} \log n, \tag{55}$$

one can show that

$$\sum_{j=1}^{n} \frac{\lambda_{1j}^2}{\left(\sum_{h=1}^{2} \theta_h \lambda_{hj}\right)^2} \leq \frac{6+3\alpha}{2b\theta_{\min}^2} \log n + \frac{C}{\theta_{\min}^2} \leq \frac{4+2\alpha}{b\theta_{\min}^2} \log n, \tag{56}$$

when $n > C$ for $C$ depending on $b$. In terms of the lower bound, first note that

$$\sum_{j=1}^{n} \frac{\lambda_{1j}^2}{\left(\sum_{h=1}^{2} \theta_h \lambda_{hj}\right)^2} \geq \sum_{j=1}^{n} \frac{\lambda_{1j}^2}{4\theta_{\max}^2 \max_h \lambda_{hj}^2}$$

$$\geq \frac{|\{j : \lambda_{1j} = \max_h \lambda_{hj}\}|}{4\theta_{\max}^2}. \tag{57}$$

Due to (53), we have

$$\lambda_{1j} = \max_h \lambda_{hj} \Leftarrow \frac{C_1}{2} n e^{-bj} \geq Cn^{1-\epsilon} + C$$

$$\Leftarrow e^{-bj} \geq Cn^{-\epsilon} \tag{58}$$

$$\Leftarrow j \leq \frac{\epsilon}{b} \log n - C,$$

when $n > C$ for some $C > 0$ depending on $b$, which implies

$$\left|\{j : \lambda_{1j} = \max_h \lambda_{hj}\}\right| \geq \frac{\epsilon}{b} \log n - C \geq \frac{\epsilon}{2b} \log n,$$

if $n > C$. Thus we have

$$\sum_{j=1}^{n} \frac{\lambda_{1j}^2}{\left(\sum_{h=1}^{2} \theta_h \lambda_{hj}\right)^2} \geq \frac{\epsilon \log n}{8b\theta_{\max}^2},$$

2. $l = l' = 2$

The upper bound for $\sum_{j=1}^{n} \frac{\lambda_{2,j}^2}{\left(\sum_{h=1}^{2} \theta_h \lambda_{hj}\right)^2}$ in Lemma 2.3 is straightforward, since $\sum_{h=1}^{2} \theta_h \lambda_{hj} \geq \theta_{\min}$. While for the lower bound, note that $\lambda_{2,j} = 1$, and thus

$$\sum_{j=1}^{n} \frac{\lambda_{2,j}^2}{\left(\sum_{h=1}^{2} \theta_h \lambda_{hj}\right)^2} \geq \frac{|\{j : \sum_{h=1}^{2} \theta_h \lambda_{hj} \leq 2\theta_{\max}\}|}{4\theta_{\max}^2}. \tag{59}$$

Meanwhile, let $\eta = \frac{3}{2}$ in (52), then one can show that

$$\sum_{h=1}^{2} \theta_h \lambda_{hj} \leq 2\theta_{\max} \Leftarrow CM\theta_{\max} n^{1+\frac{\alpha}{2}} e^{-\frac{bj}{3}} \leq \theta_{\max}$$

$$\Leftarrow j \geq \frac{6+3\alpha}{2b} \log n + C \tag{60}$$

$$\Leftarrow j \geq \frac{6+3\alpha}{b} \log n$$

when $n > C$ for $C > 0$ depending on $b$. Therefore,

$$\sum_{j=1}^{n} \frac{\lambda_{2,j}^2}{\left(\sum_{h=1}^{2} \theta_h \lambda_{hj}\right)^2} \geq \frac{n - \frac{6+3\alpha}{b} \log n}{4\theta_{\max}^2}. \tag{61}$$

3. $l = 1, l' = 2$

   First note that by similar arguments from the first case where $l = l' = 1$, one can show that

   $$\frac{\lambda_{1j} \lambda_{2,j}}{\left(\sum_{h=1}^{2} \theta_h \lambda_{hj}\right)^2} \leq \min\left\{\frac{1}{4\theta_{\min}^2}, \frac{C(\eta) n^{1+\frac{\alpha}{2}} e^{-\frac{b}{2\eta} j}}{\theta_{\min}^2}\right\}, \tag{62}$$

   one can show that

   $$\sum_{j=1}^{n} \frac{\lambda_{1j} \lambda_{2,j}}{\left(\sum_{h=1}^{2} \theta_h \lambda_{hj}\right)^2} \leq \frac{(2+\alpha)\eta \log n}{4b\theta_{\min}^2} + \frac{C(\eta) n^{1+\frac{\alpha}{2}}}{\theta_{\min}^2} \sum_{j=\lceil \frac{(2+\alpha)\eta}{b} \log n \rceil}^{n} e^{-\frac{b}{2\eta} j}$$

   $$\leq \left(\frac{(2+\alpha)\eta}{4b} + \frac{C(\eta)}{\log n}\right) \frac{\log n}{\theta_{\min}^2}. \tag{63}$$

   Let $\eta = \frac{8}{7}$, then when $n > C$ for some $C > 0$ depending on $b$,

   $$\sum_{j=1}^{n} \frac{\lambda_{1j} \lambda_{2j}}{\left(\sum_{h=1}^{2} \theta_h \lambda_{hj}\right)^2} \leq \frac{(5+2\alpha) \log n}{7b\theta_{\min}^2}.$$

Therefore, for any $0 < \epsilon, \alpha < 1$, if $n > C(\epsilon)$ for $C(\epsilon) > 0$ depending on $b, \epsilon$, then with probability at least $1 - 2n^{-\alpha}$, (15) holds. $\qquad\square$

*Proof of Lemma 3.1.* First note that

$$\left| z_n^\top \boldsymbol{\Lambda}^{(H)}(\boldsymbol{\theta}) z_n - \text{tr}(\boldsymbol{\Lambda}^{(H)}(\boldsymbol{\theta})) \right| = \left| \sum_{j=1}^{n} \boldsymbol{\Lambda}_{jj}^{(H)}(\boldsymbol{\theta})(z_{nj}^2 - 1) \right|$$

$$\leq \sum_{j=1}^{n} \boldsymbol{\Lambda}_{jj}^{(H)}(\boldsymbol{\theta}) |z_{nj}^2 - 1|. \tag{64}$$

By the definition of $\boldsymbol{\Lambda}^{(H)}(\boldsymbol{\theta})$, $\varepsilon$, $t_i(n)$, (37) and (38),

$$\|\boldsymbol{\Lambda}^{(H)}(\boldsymbol{\theta})\|_2 \leq \|\boldsymbol{\Lambda}(\widetilde{\boldsymbol{\theta}})\|_2 (H+1) \left(\frac{2\varepsilon}{\theta_{\min}}\right)^H \leq Ce^{-H},$$

$$\|\boldsymbol{\Lambda}^{(H)}(\boldsymbol{\theta})\|_F^2 \leq e^{-2H}\|\boldsymbol{\Lambda}(\widetilde{\boldsymbol{\theta}})\|_F^2 \leq Ce^{-2H}t_i(n).$$

Also note that following similar arguments for bounding $\|\boldsymbol{\Lambda}(\boldsymbol{\theta}_\varepsilon^{(k)})\|_F^2$, we have

$$\sum_{j=1}^n \left|\boldsymbol{\Lambda}_{jj}(\widetilde{\boldsymbol{\theta}})\right| \leq Ct_i(n), \tag{65}$$

and thus

$$\sum_{j=1}^n \left|\boldsymbol{\Lambda}_{jj}^{(H)}(\boldsymbol{\theta})\right| \leq e^{-H} \sum_{j=1}^n \left|\boldsymbol{\Lambda}_{jj}(\widetilde{\boldsymbol{\theta}})\right| \leq Ce^{-H}t_i(n). \tag{66}$$

Therefore,

$$\mathbb{P}\left(\left|z_n^\top \boldsymbol{\Lambda}^{(H)}(\boldsymbol{\theta})z_n - \mathrm{tr}(\boldsymbol{\Lambda}^{(H)}(\boldsymbol{\theta}))\right| > e^{-H}\left(t + Ct_i(n)\right)\right)$$
$$\leq \mathbb{P}\left(\sum_{i=1}^n \boldsymbol{\Lambda}_{jj}^{(H)}(\boldsymbol{\theta})(|z_{nj}^2 - 1| - \mathbb{E}(|z_{nj}^2 - 1|)) > e^{-H}t\right), \tag{67}$$

where $C > 0$ depends on $\theta_{\min}, \theta_{\max}, b$. Since $|z_{nj}^2 - 1|$ is sub-exponential with constant parameter,

$$\mathbb{P}\left(\sum_{j=1}^n \boldsymbol{\Lambda}_{jj}^{(H)}(\boldsymbol{\theta})\left(|z_{nj}^2 - 1| - \mathbb{E}|z_{nj}^2 - 1|\right) > e^{-H}t\right) \leq 2\exp\left\{-c\min\left\{\frac{t^2}{t_i(n)}, t\right\}\right\}. \tag{68}$$

$\square$

# 4    Additional Figures in Numerical Studies

Under the setup described in Section 6.1, we also investigate how minibatch size $m$ influences the convergence of parameters, as shown in Fig. 4.1. We see that larger minibatch size results in faster convergence and smaller statistical error (more concentrated curves) for the parameters.

Figure 4.1: Comparison of the convergence of parameters with varying minibatch sizes. Lines in black denote the true parameters. The three experiments share initial point $\boldsymbol{\theta}^{(0)} = (5.0, 3.0)$ and inital step size $\alpha_1 = 9$.