[Reviews · NeurIPS 2020]

Review 1

Summary and Contributions: A minibatch estimate of the loss used to train GPs (the marginal log likelihood) corresponds to training with artificially imposed block diagonal structure on the prior covariance matrix. While minibatch training procedures may work empirically, to the best of my knowledge there has to date been no known theoretical connection suggesting that one might recover anything close to the true optimal hyperparameters. In this paper, the authors resolve this issue by providing (under only one assumption that I would view as limiting) precisely such a theoretical connection.

Strengths: This paper provides a strong theoretical result justifying the use of SGD for training exact GPs. Even in the limited setting considered (e.g., the theory mostly applying to RBF kernels), this is in my opinion a strong accomplishment.

Weaknesses: I have a few minor detailed comments about how the empirical evaluation could be improved: see the "Correctness" and detailed comments boxes below.

Correctness: The theoretical results appear correct, although as the authors point out, the theory is limited to kernels with very strong eigenvalue decay rates. Given that the theory as presented precludes e.g. Matern kernels, it would be nice to see a simple empirical evaluation on these kernels. I do believe the evaluation in Table 1 is somewhat flawed; see my second paragraph in the detailed comments section below.

Clarity: The paper is clearly written, and relatively free of typos.

Relation to Prior Work: Clearly, yes. While applying SGD directly to the exact GP NLL may have been tried before, I'm not aware of any paper or framework under which it would have been viewed as a correct or sensible thing to do prior to this work. The one comment I'll make about related work is that, at least at this point, techniques like SVI for GPs do not exist solely to accelerate GP regression anymore. They enable not only non Gaussian likelihoods (e.g., for classification), but also much more complex classes of models like deep GPs.

Reproducibility: Yes

Additional Feedback: I'd like to see main paper Figure 1 / supplementary figure 4.1 expanded. The two questions I have that I don't think the figure currently answers are (1) how does the variance in final \sigma^{2}_{f} across trials compare to a full batch GP, and (2) if full batch GPs have smaller variance, do much larger batch sizes (e.g., say m=1000) decrease this variance further? In figure 4.1, it does not seem the variance decreases much from m=16 to m=64 -- it'd be nice to know whether the batch size is the source of the variance. If it is, then running with very large batch sizes even up to m=10000 may not be too challenging. To the point of running large batch sizes, while the ability to use SGD will clearly outperform full batch training at some size N (at a guess, probably somewhere in the the N=100k-500k range), I don't think the results in Table 1 are necessarily representative of the settings you might actually want to run sgGP or EGP with. Training on a CPU masks the fact that sgGP is performing a very large number of sequential operations relative to EGP, and the methods you are using for EGP are pretty terribly slow on a CPU. For example, you list Protein as taking 500 minutes to train on; with access to a single consumer grade GPU, this actually takes about 70 seconds. Given that your training procedure will involve processing ~170000 minibatches for this dataset, sgGP would almost certainly still be slower in this regime. However, I could easily imagine sgGP again being faster with GPU acceleration using minibatch sizes on the order of 10000. ---------- My opinion of this paper did not change during the discussion -- I quite liked it and attempted to champion it during the discussion phase. As I mentioned in the discussion, several of the reasons raised to reject the paper I simply found to be unconvincing or wrong. I believe that this paper should be accepted. I will note, however, that I found myself in agreement with the other reviewers that the M>1 case should simply be removed from the paper. The assumption required to get the theory working in this case will virtually never hold in practice, particularly if you are using different lengthscales in practice. Therefore, the authors should restrict themselves to the M=1 case, where the results are clear and not muddied by an overly strong assumption that adds very little to the paper.


Review 2

Summary and Contributions: This paper suggests using stochastic gradient descent (SGD) for model selection in Gaussian Processes (GPs) for regression tasks. SGD is applied to the negative log marginal likelihood with the ultimate goal of improving computational complexity. Indeed, optimizing the negative log marginal likelihood requires the inversion of the matrix of the GP kernel plus the Gaussian observational noise, which scales as $O(n^3)$, where $n$ is the number of data samples. Using SGD allows computing the gradient of the negative log marginal likelihood on a mini-batch of size $m<<n$ of the data points, which can dramatically speed up inverse computation with traditional techniques such as the Cholesky decomposition. The main contribution of this work is to show that, under some assumptions, SGD converges to the global minimum of the loss function, intended as the negative log marginal likelihood. Indeed, under some assumptions, the authors show that the SGD iterates converge at a rate $O(1/K)$, and that with sufficiently many iterations and a large enough mini-batch size, the SGD converges to a critical point of the loss. All results rely on the strong convex-like property of the SGD process, which derives from assuming kernel matrices to exhibit exponential eigendecay, as proved in the literature. Some numerical experiments complement the theoretical contribution of this work. Authors compare SGD-based model selection to alternative methods from the literature, including exact GP methods designed to scale to distributed settings, as well as to approximate GPs based on inducing points (model approximation) or on stochastic variational inference (inference approximation).

Strengths: The paper addresses an important topic for the GP community, that is how to make GP model selection (and possibly GP inference) scalable and computationally efficient. - The main contribution of this work is theoretical: under some assumptions, they prove that using the wide-spread SGD algorithm on mini-batches of input samples can converge to the local minimum of the loss, intended as the negative log marginal likelihood, which is used to optimize the hyperparameters of a GP - Although it is not very clear from the theory, they experimentally show that their approach dramatically outperforms, computation time wise, the model selection procedure (what they call "training") of a GP, which translates into better performance in terms of RMSE on the test set, supposedly because better hyperparameters are found by the proposed method

Weaknesses: - positioning of this work: I think the authors could improve their paper by polishing the way they position their work w.r.t. the literature. Although as a reviewer I am familiar with the GP literature and the cornerstone books such as Rasmussen and WIlliams, I had some difficulties in fully grasping the contribution of this work. Most of the related work deal with the inference problem, that is how to avoid or mitigate paying the $O(n^3)$ cost related to GP regression. Indeed, in GP regression, we have analytical expressions for the distribution of the latent functions, and for the predictive distributions (their mean and covariances). So, there is actually no "training" in the common way it is intended: one need to perform matrix inversions that are extremely costly. So, the literature focused either on model approximations (inducing points, random Fourier features) or on inference approximations (variational inference) or on both. Generally, GP regression hyperparameter optimization (model selection) is glossed over. In this work, instead, model selection takes a prime spot, but what is not clear is how the proposed SGD technique carries over to prediction tasks. Indeed, text book material indicate (see chapter 5 of Rasmussen and Williams) that once we have the inverse of the kernel times the output, computing the posterior predictive distribution can be dramatically simpler, because of computation reuse. Does this apply to the proposed method? - numerical experiments: I think the authors miss an opportunity in the first part of their synthetic data experiments, where they could have investigated more whether the assumptions they rely on for the theoretical work actually hold in practice. In particular, I am curious to see if assumptions 3.1 and 3.3 hold. In the case studies, I think the authors are a bit unfair to the competitor EGP, which was designed to work on a distributed setup with multiple GPUs. Indeed, they perform all experiments on a single CPU cure (to the best of my understanding). It would have also been very useful trying to explain why alternative methods seem to perform worse than the proposed approach: is it because they are approximate, and as such are doomed to find sub-optimal hyperparameters for the GP?

Correctness: To the best of my understanding, the claims and methods in this work are correct. Most of the material is based on text book formulations (chapter 5 from Rasmussen and Williams). The original work is to show that SG of the loss enjoy properties such that they converge to the minimizers, given some assumptions on the GP kernels. Also the empirical methodology appears to be mostly correct, although in one case, to the best of my understanding, the comparison to a competitor seems to be a bit unfair, as it was designed to work on multiple GPUs, whereas the experiments are run on a single CPU.

Clarity: Yes, the paper is clearly written, with a couple of typos here and there that are easy to spot. Maybe, as a constructive advice to the authors, I could suggest the following: save space by compressing the text-book material from section 2, and dedicate more space to proof overview. This is the main contribution of the paper which, even if it relies on existing literature, should be given more space. One thing that could be discussed more is the validity of the hypotheses (3.1 and 3.3), especially in practical endeavors.

Relation to Prior Work: I think the related work section could also enjoy some polishing. The literature review on exact, sparse, and approximate inference for GP regression is mostly well covered. Instead, the literature on model selection is less discussed: one suggestion could be to have a look at papers such as: Stefan Falkner, Aaron Klein, and Frank Hutter. 2018. BOHB: Robust and Efficient Hyperparameter Optimization at Scale. In Proceedings of the 35th International Conference on Machine Learning (Proceedings of Machine Learning Research), Jennifer Dy and Andreas Krause (Eds.), Vol. 80. PMLR, Stockholmsmassan, Stockholm Sweden, 1437–1446.

Reproducibility: Yes

Additional Feedback: General comments: in addition to all the points above, here are some general comments. As a general remark, I think the introduction should clarify/stress the relation between the quest for exact GP inference and GP model selection. Does it make sense to perform model selection on approximate models? I think that Cutajar [ICML 2017, Deep Gaussian Processes with Random Fourier Feature] does this using the approximate marginal log likelihood. I think it would be good to discuss what do we gain from *exact* model selection? In the experiments of your paper, you somehow show that the hyper-parameter you obtain are substantially better than an approximate model selection strategy, so maybe this could be discussed already in the introduction. Also, on another note, in the introduction we have the following sentence: <<we establish convergence guarantees for both the full gradient and the model parameters. Interestingly, without convexity or even Liptchitz conditions on the loss function, ... >> In reality, the assumptions are not explicit, but implicit, as they derive from the hypotheses on the kernel properties, that is the exponential decay of the eigenspectrum of the kernels. - Section 2: line 92, 93: Why this assumption? Do you need this for your work? In standard 101 GP formulations you have a simple kernel function, not a linear combination of $M$ kernels. I think it should be stated explicitly you only focus on model selection for GP regression, and maybe cite the Rasmussen and Williams book on GP for expressions (3) and (4). In section 2.1, crucially, I think you should stress more the idea that mini-batches of input samples used to produce the covariance $K_{\xi}$ make it so that this matrix is more efficiently invertible when $m<<n$, so your complexity goes down to $O(m^3)$. In algorithm 1: If there is a learning rate decay schedule in the algorithm, how can you make sure the learning rate remains within the assumed bounds, that by the way depend on the largest parameter, which means that the values of \alpha could be pretty small? This is related to the bounded iterates assumption. - Section 3: * I think it would be helpful to clarify from the outset that we will seek at finding "good" properties for SGD to converge, such as equivalent convexity properties that will help guarantee convergence * Is there any citation you could mention that could support the assumptions 3.1 and 3.3? - Section 4: * I couldn’t verify everything, but the arguments used in this section seem to go well with the intuition and the role of the hypotheses. The only thing I am missing is a reference to explain that these assumptions actually do hold in practice. Then, everything else looks fine to me. * I find the notation "*" used here is a bit confusing: it used to indicate "optimality" in section 3, now it is used to indicate a conditional expectation * Lemma 4.2: This is a fundamental lemma, on which all your work is based on. It follows from the assumption on the egindecay of the kernel matrices, for otherwise, in general, the loss could have multiple local minima. This is exacerbated with fewer data points, whereas when $n$ is sufficiently large, one local minima can become extremely more probable than others. However, it might be the case that you have a situation like in fig. 5.5 of Rasmussen and Williams, and you rule out this by making “peculiar” assumption on the kernel properties. So again, are those assumptions generally valid? - Section 5: * I think here we miss some other approximation methods, where both the GP model is approximate, e.g. using random Fourier feature, and the inference is approximate using VI. Overall though, related work seem to be more concerned about the applicability of GPs to solve real world problems with large dataset sizes. By no means they look at model selection, which is the core topic of this work. It might be relevant to look at other work including multi-fidelity GPs that also treat model selection (see reference in previous comments). - Section 6: * 6.1: For these numerical toy regression examples, it would have been interesting to show if assumptions 3.1 and 3.3 hold in practice. Indeed, these are fundamental assumptions for the strong convexity of the loss, which help SGD reach a global optimum. * Fig 2: This is just an observation: how come when you increase the mini batch size the behavior of the slopes is more jiggly? I would expect the other way around! * 6.2: I’m not sure I understand why you need to compare your method to exact and sparse GP regression. You aim at optimizing hyper-parameters of the GP, not at computing the predictive distribution right? So what exactly are we comparing here? * Is the comparison to EGP fair? EGP was conceived mainly to work on multiple GPUs, whereas here you seem to focus on a single CPU. Can you clarify? =========== Additional comments based on the rebuttal from the authors, and the reviewers' discussion =========== Although I was already leaning toward an accept for this paper, author's rebuttal and reviewers' comments really helped clarifying some of my points. Overall, it seems that the theoretical contribution outweighs the weaknesses of this work. * For Assumption 3.1, this is satisfied for RBF kernels. For kernels with a polynomial decay rate (e.g. Matern), the authors claim to have additional theoretical results. I strongly suggest to add these results in the eventual camera ready * I tend to agree with other reviewers that ditching the case for M>1 would really help clarify this work. Then if Assumption 3.3 would not be needed anymore, the repercussions on the clarity of the paper would be important and make it simpler to parse * Algorithm 1 should be amended with the additional detail on picking the nearest neighbors (which is not a "free" operation in terms of costs), and experiments should report results with and without this additional technique As a consequence, I have updated my score from 6 to 7


Review 3

Summary and Contributions: The paper proves convergence of minibatch SGD methods when learning a GP formed by a sum of kernel (covariance) functions.

Strengths: Convergence guarantees for learning a GP via SGD (or other methods) are notoriously difficult. While I wonder about the formulation and conditions (see below) the work here is a good start.

Weaknesses: It appears there is a disconnect between the theory and the results. The parameters considered in the theory (see line 96) are all scale parameters and do not include the lengthscale parameters. I also wondered about Assumption 3.3 (line 135). As the authors acknowledge 2 lines later, it is a very strong assumption. But the consequence is not spelled out. If the kernel matrices share the same eigenvalues, does this mean they share some aspects of the correlation structure? I was particularly confused by the comment that the assumption is not necessary for M = 1: in that case can't the single scale parameter be optimized in closed form for both maximum likelihood and common Bayes formulations? The illustration in Section 6.1 confirms that the lengthscale has to be known. The case studies in Section 6.2 do introduce learning of the different lengthscales. When this is done, I doubt the theory still applies. When the conditions of the theory do not apply, it is not clear to me how Algorithm 1 differs from other minibatch methods and why it should beat them in the Results section. The role of correlation, as in the title, is not clear to me. See Remark 3.2 (line 154) and the comment "the fact that statistical errors depend on m instead of n is due to the correlation". Why does correlation have a different effect on m (the minibatch size) compared with m? ADDED after authors' feedback: You are correct that estimating the two variances introduces essentially one correlation parameter (the variances' relative sizes) and one overall scale parameter. The latter can be estimated in closed form (max likelihood or Bayes) so we basically have a 1-d optimization problem of the single correlation parameter. I still think this is limiting as with strong signal (the case of practical interest) the length scale parameters are really important.

Correctness: The claims appear to be correct, but I wonder whether their (honestly stated) conditions render them impractical.

Clarity: The paper is well structured and free of typos. K appears in the abstract and many times before its definition (I think) in Algorithm 1 on p.4. This is particularly unfortunate when bold K is also used for a different quantity.

Relation to Prior Work: It was not clear to me how this differs from other SGD methods when we get to the real applications and more parameters are learned.

Reproducibility: Yes

Additional Feedback:


Review 4

Summary and Contributions: The authors derive and present a set of theoretical guarantees for estimating Gaussian processes using minibatch stochastic gradient descent. Under certain assumptions on the gradients, parameters, and kernel, the authors demonstrate that the SGD estimate of the variance of the additive noise term and the signal variance of the kernel with the slowest eigendecay converge to the true parameters with optimization error bound of O(1/K) and a statistical error bound of O(m^(-1/2)) for the noise variance and of O((log m)^(-1/2)) for this signal variance, where K is the number of iterations and m is the size of the minibatch. Additionally, the authors prove the minibatch SGD estimate of the gradient converges to the full gradient with O(1/K) optimization error and O(m^(-1/2)) statistical error. These results are proved without convexity or Lipschitz conditions on the loss function. Finally, the authors apply the SGD estimation technique to a battery of synthetic and real benchmark datasets showing that it compares favourably with other leading estimation techniques in terms of RMSE and massively outperforming them in terms of computation time and and storage space.

Strengths: The strengths of the paper are as follows: 1) The subject areas, efficient estimation of GPs and deriving theoretical guarantees for SGD, are clearly of great interest and importance to the NeurIPS community, and getting a better grasp of the theoretical guarantees of SGD in such settings is certainly a promising area of research. 2) The theoretical claims are sound and are well documented, giving a good perspective of how such claims can be proven in this setting. 3) The empirical results demonstrate that minibatch SGD can improve on existing techniques, substantially in some cases, in terms of RMSE, and the massive reduction in time and space complexity certainly opens new application areas for GPs.

Weaknesses: The paper has several weaknesses that must be addressed or clarified, most of which are about the generality of the results: 1) While there is no restriction on the loss function, it does appear that the results are limited to regression tasks as suggested by section 2 and the empirical results section. As many of the competing estimation techniques have been developed or extended with classification tasks in mind, the paper would be strengthened by an inclusion of how the results translate to this case both theoretically and empirically. 2) The restrictions on the form of the kernel functions Assumptions 3.1 and 3.3 in conjunction seem severe, it would be good to have a description of what families of kernel functions satisfy these constraints together. 3) The parameters of focus for the theoretical analysis are only the signal variances, this further restricts the expressibility of the resulting processes in addition to point (2). Furthermore, the scope of the theoretical results is constrained to two parameters: the noise variance and the signal variance of the kernel with the slowest eigendecay. It would of course be nice if the results could be extended to convergence guarantees for all the parameters, but a discussion of why this limitation occurs would be elucidating. 4) The empirical results section mentions that the minibatch SGD algorithm is augmented by forming a minibatch by drawing a random datapoint and finding its 15 nearest neighbours. I think this is an acceptable addition to the method, but I would like to see results using only Algorithm 1. This would give a fuller picture. My concern is that the combination of all these points (limited to regression tasks, restrictions on expressibility, and a helping hand in finding the gradient by constructing nice minibatches) leads to a perfect setting for SGD to perform well relative to the other methods. Addressing these points would make the analysis more robust and the paper stronger.

Correctness: The theoretical results as presented in the paper and appendices are correct, and the empirical methods are correct, barring the discussion above.

Clarity: The paper is mostly well written, but some points: 1) Eq. 2 and line 98 seem to contradict line 248, are the parameters under consideration only the signal and noise variances, or do they include other parameters of the kernel functions? 2) The equation in line 124 is a little unclear as written, it could be clearer as l \in {1, M+1} or with the inclusion of an ‘or’. 3) Line 233 claims the curves become ‘less concentrated’ with minibatch size in Figure 2, a hard number would be useful here. 4) Line 256 reads ‘nearest neighborhoods’, this should be neighbours.

Relation to Prior Work: The paper positions itself well in terms of previously proposed estimation techniques such as matrix vector multiplication, sparse approximate inference, and stochastic variational inference methods as documented in the introduction and Section 5. However, there is a noticeable lack of discussion of other attempts to bring SGD to bear on GPs, such as [Yan, Xinyan, Xie, Bo, Song, Le, and Boots, Byron. Large-scale Gaussian process regression via doubly stochastic gradient descent. The ICML Workshop on Large-Scale Kernel Learning,2015.]

Reproducibility: Yes

Additional Feedback: Update after rebuttal and discussion: The rebuttal and discussion cleared up a few problems with the paper and I am revising my review up. However, I still think the paper needs some serious changes: 1) Cut the case in which M > 1. The consensus seems to be that this does not add to the discussion and muddies the waters as to the actual contribution of the paper by requiring Assumption 3.3. 2) Justify with citations or remove lines 274-6 ‘our results demonstrate the implicit regularization effect by utilizing correlated minibatches, i.e. the algorithm tends to approach local minimas that have better generalization performance’. This shifts the claim of the paper from “SGD is a way to estimate GPs, and now have strong theoretical guarantees on some hyperparameters (a good, clear, and strong presentation of the results)” to “SGD is THE way to estimate GPs (much less justified and oversold reading of the results)”. 3) Include results without the nearest neighbour search. This search is not costless and is not discussed elsewhere in the paper.

[Author Response · NeurIPS 2020]

We thank all reviewers for their careful reading and useful comments. Due to space limit, we focus our response on the main comments. In particular, we notice Assumption 3.1 and Assumption 3.3 drew most attentions. *We'd first like to state that despite these assumptions, to the best of our knowledge, this paper presents the first successful try at proving SGD converges (and recovers the true noise) in correlated settings. Open problems still exist, some assumptions can be further relaxed, yet we believe this paper presents the first crack at this challenging topic.*

1. **Explanation on Assumption 3.1:** We would like to point out that the exponential eigendecay assumption (Assumption 3.1) is satisfied by a wide range of kernels including the RBF kernel (the Gaussian kernel), see Section 4.3.1 in Rasmussen, C. E. (2003), "Gaussian processes in machine learning", which is commonly seen in the GP literature. Other kernels with non-exponential eigendecay mostly decay at a polynomial rate, e.g., Matern kernel, see section 2.3 in Bach, F. (2017), "On the equivalence between kernel quadrature rules and random feature expansions". We do have additional theoretical guarantees for kernels with polynomial eigendecay where the optimization error is still $O(\frac{1}{K})$ and the statistical error is $O(m^{-\frac{1}{2}+\varepsilon})$ where $0 < \varepsilon < \frac{1}{2}$ depends on the particular decay rate. We will include these results in the camera-ready version if accepted.

2. **Explanation on Assumption 3.3:** When $M = 1$, the considered model is the standard GP formulation and satisfies Assumption 3.3 directly. Meanwhile, in response to reviewer 3's question, both signal variance and noise variance need to be estimated when $M = 1$, and there is no closed form solution for them together. When extending $M = 1$ to $M > 1$, the already challenging proof becomes extremely challenging without Assumption 3.3. We also have simulation results which were not included due to space constraints, suggesting that SGD works well for $M > 1$. Providing theoretical guarantees for $M > 1$ without Assumption 3.3 remains an open and challenging question.

3. **Estimation for lengthscale:** To clarify, we only claim the convergence guarantee for estimating signal and noise variances. The extension to convergence guarantees for the lengthscale parameter in RBF kernel is extremely challenging: both the proof for Lemma 4.1 and 4.2 presents additional challenges if looking at the lengthscale since it lies inside the exponential as a denominator. However, the case studies suggest that SGD may still be used for estimating the lengthscale in practice. Meanwhile, we also have additional simulation experiments suggesting that SGD also recovers the true lengthscale up to some statistical error, which will be added if space permits.

4. **Theoretical contribution:** We also want to emphasize the technical challenges even when we consider estimating only the signal and noise variances.

   First we explain the role of "correlation": here we mean the correlation of $\{\mathbf{y}_{\xi_k}\}_{k=1}^K$ conditioning on $\mathbf{X}$, which results in the correlation of $\{g(\boldsymbol{\theta}^{(k-1)})\}_{k=1}^K$. The statistical error is the deviation of $g(\boldsymbol{\theta}^{(k-1)})$ from $\mathbb{E}(g(\boldsymbol{\theta}^{(k-1)})|\mathbf{X}_{\xi_k})$, averaged over each iteration $1 \leq k \leq K$, which is less concentrated (larger variance) if the correlation among $g(\boldsymbol{\theta}^{(k-1)})$ is strong. Therefore the statistical error only converges to 0 when the minibatch size $m$ tends to $\infty$, thus $m$ is assumed to be large instead of being a constant.

   In order to have a lower bound on the approximate curvature (see Lemma 4.2) that is independent of $m$, we establish novel upper and lower bounds on $\sum_{j=1}^m \lambda_{lj}\lambda_{l'j}(\sum_{i=1}^M \theta_i^{(k)}\lambda_{ij} + \theta_{M+1}^{(k)})^{-2}$ with high probability when $m$ is large, where $\lambda_{lj}$ is the $j$th eigenvalue of $\mathbf{K}_{f,n}^{(l)}$.

   The proof for Lemma 4.1 is also non-trivial since the error bounds holds uniformly for all $\boldsymbol{\theta} \in [\theta_{\min}, \theta_{\max}]^{M+1}$, and the gradient involves the trace of an $m \times m$ high-dimensional matrix whose entries all depend on $\boldsymbol{\theta}$ in a non-linear way. We apply Taylor's expansion and a novel truncation technique to avoid the difficulty caused by the high non-linearity.

5. **Model selection v.s. model inference/prediction:** The model selection (estimation of hyperparameter) is indeed our core task in the paper, which is also an important problem in GPs. The case studies presented in this paper demonstrate that SGD helps us find better hyperparameters and thus has better prediction performance. The proposed SGD technique cannot directly carry over to the prediction task, but existing methods on prediction can all be applied after the model selection. One should note that for prediction we only need to invert the kernel matrix once, but for model selection, each update of hyperparameters requires one inversion, which is more time-consuming. The speed-up of training process provided by SGD still leads to huge overall computational savings.

6. **Comparison with EGP (fairness):** For data size of $10^6$, EGP took a few days to fit with 8 GPUs while our model took 30 minutes to train on 1 CPU. We note that kernel matrix partitioning done via GPU in EGP is for acceleration and not enhanced performance over exact inference. Numerical instabilities are addressed via the kernel matrix preconditioner–a task that we also do when comparing with EGP. In addition, although SGD in correlated setting is a sequential operation, for minibatch size over $10^4$, GPU acceleration can be achieved in similar fashion as EGP via kernel matrix partitioning.

[Meta-Review · NeurIPS 2020]

All the reviewers agree that the paper presents a worthwhile theoretical contribution, which may facilitate/motivate further work to tackle more challenging problems. The main limitation of the work is its practical impact as the proposed analysis does not apply to the lengthscales. Although R3 stands by their comments, they expressed their willingness to accept and recognized, during discussions, this work as an excellent attempt at the problem. Overall, I believe the NeurIPS community will benefit from this work and recommend the authors to take the reviewers' suggestions and comments into consideration.